# ChronoEpilogi: Scalable Time-Series Variable Selection with Multiple Solutions

**Etienne Vareille**[1]  **Michele Linardi**[1]  **Ioannis Tsamardinos**[2]  **Vassilis Christophides**[1]

[1]ETIS UMR-8051 Laboratory, CY Cergy Paris Universite, ENSEA, CNRS
[2]Computer Science Department, University of Crete, Heraklion, Greece

## Abstract

We consider the problem of selecting all the minimal-size subsets of multivariate time-series (TS) variables whose past leads to an optimal predictive model for the future (forecasting) of a given target variable (multiple feature selection problem for times-series). Identifying these subsets leads to gaining insights, domain intuition, and a better understanding of the data-generating mechanism; it is often the first step in causal modeling. While identifying a single solution to the feature selection problem suffices for forecasting purposes, identifying all such minimal-size, optimally predictive subsets is necessary for knowledge discovery and important to avoid misleading a practitioner. We develop the theory of multiple feature selection for time-series data, propose the ChronoEpilogi algorithm, and prove its soundness and completeness under two mild, broad, non-parametric distributional assumptions, namely *Compositionality* of the distribution and *Interchangeability* of time-series variables in solutions. Experiments on synthetic and real datasets demonstrate the scalability of ChronoEpilogi to hundreds of TS variables and its efficacy in identifying multiple solutions. In the real datasets, ChronoEpilogi is shown to reduce the number of TS variables by 96% (on average) by conserving or even improving forecasting performance. Furthermore, it is on par with Group Lasso performance, with the added benefit of providing multiple solutions.

## 1   Introduction

In analysis tasks involving TS with hundreds or even tens of thousands of variables (e.g., in manufacturing, environmental monitoring, energy grids, etc.), selecting appropriate series for TS forecasting not only has the potential to improve models' performance, but also to gain intuition, discover new knowledge, and understand the data-generating mechanism (causal structure). *Time-series variable selection* (TVS) is defined as the problem of discovering *a minimal-size subset of the available, measured TS variables whose past values optimally predict the future of a target TS variable $T$*. Such a series subset is called **a Markov Boundary** (MB) of $T$ in the causal discovery literature [Pea88], denoted as $\mathbf{MB}(\mathbf{T})$. Hence, $\mathbf{MB}(T)$ filters out the TS variables that are both *irrelevant* and *redundant* in forecasting $T$.

Identifying a small-size $\mathbf{MB}$ leads to a computationally more efficient model for $T$ when real-time predictions are required. Moreover, it facilitates all subsequent analysis operations such as modeling, explanation calculations (e.g., SHAP values, feature importance), visualizations, and interpretations [VALC23]. Perhaps most importantly, identifying the $\mathbf{MB}$ of every series could be the first step in *causal modeling* of the data-generating mechanism. Indeed, under broad causal assumptions, the $\mathbf{MB}(T)$ contains the observable direct causes of $T$, i.e., the quantities that could causally influence and optimize the future values of $T$ [LTB$^+$16]. It is not an accident then that variable selection is the first step in several causal discovery algorithms for time-series or cohort data [AST$^+$10b, AST$^+$10a]. However, we note that the $\mathbf{MB}(T)$ may also contain variables confounded by latent ones (in general, variables connected to $T$ by a collider path). Distilling the possible direct causes of $T$ out of its

$\mathbf{MB}(T)$ requires further analysis with causal discovery algorithms, which is outside the scope of our work (readers are referred to [ADG22] for a recent survey).

Both in theory and in practice, there may be *numerous Markov Boundaries of $T$*. This is, for example, the case if a series subset $\{X, Y, Z\}$ is 1-to-1 deterministically related with another subset $\{A, B\}$, in which case, either subset could substitute for the other in an optimal forecasting model. We call such subsets **informationally equivalent series**. Equivalent series define an equivalence class. The total number of Markov Boundaries of $T$ is exponential to the number of equivalence classes to which the members of $\mathbf{MB}(T)$ belong. This is because we can construct a new $\mathbf{MB}(T)$ by picking any of its subsets and substituting it with an equivalent one from its equivalence class. Determinism is not the only reason for multiple MBs in practice: when the sample size is finite it may be impossible to distinguish the true $\mathbf{MB}(T)$ from some other subset that leads to a forecasting model of statistically indistinguishable performance. The presence of an (often exponential) number of multiple MBs has been established in cohort, cross-sectional data [TCP$^+$22, BT21, SLLA13]. For time-series data there have not been any algorithms that return multiple solutions to the TVS problem [YLL21, SLL$^+$15].

Identifying all subsets $\mathbf{MB}(T)$ (referred to as the *multiple TVS (MTVS) problem*) is crucial. While finding any one $\mathbf{MB}(T)$ suffices for forecasting, this is insufficient for knowledge discovery and model interpretation. A practitioner may find it misleading to construct an optimal and minimal-size forecasting model by filtering out certain series although there exists another $\mathbf{MB}(T)$ that contains them. In essence, multiple $\mathbf{MB}(T)$s indicate the presence of multiple causal models and explanations that fit the data equally well. Moreover, if two series $X$ and $Y$ are informationally equivalent w.r.t. forecasting $T$, a variable selection algorithm may inconsistently select between them during cross-validation or bootstrapping - arbitrarily returning either $X$ or $Y$ at each step. Feature Importance XAI methods may suffer from similar instability, on top of already known misleading interpretations. For instance, Shapley values currently suffer from the inclusion of unrealistic data instances when features are correlated, even for linear models [AJL21].

In this paper, we define the novel problem of *multiple time-series variable selection* (MTVS), we introduce the concepts of TS informational equivalence and provide a taxonomy to characterize TS variables w.r.t. to their presence in multiple $\mathbf{MB}(T)$: *indispensable*, variables that belong in all $\mathbf{MB}(T)$, *replaceable*, variables that belong in some $\mathbf{MB}(T)$ but could be replaced by some other series in the same equivalence class, *redundant*, variables that are informative but do not belong in any $\mathbf{MB}(T)$, and *irrelevant*, variables that are completely uninformative. We describe the property of *Compositionality* of the data distribution that allows the construction of *sound* yet greedy MTVS algorithms. Additionally, we define the property of *Interchangeability*, which specifies that any two $\mathbf{MB}(T)$s can be decomposed into pairs of equivalent single variables. Under the assumption of Interchangeability, one variable can be substituted with another variable to create a new MB. This property enables the design of algorithms that are *complete* in identifying all MBs without requiring exhaustive enumeration. This is particularly valuable in data distributions with an exponential number of MBs.

We proceed with designing a MTVS algorithm, named **ChronoEpilogi**[1][2] that is sound and complete in terms of identifying and returning all MBs of $T$, under Composition, Interchangeability, and other broad, non-parametric assumptions. ChronoEpilogi extends previous variable selection algorithms designed for cohort data [BT21] to a TS forecasting setting and the identification of multiple solutions (MBs). It returns a reference $\mathbf{MB}(T)$ and a list of equivalence classes. Each of the classes contains TS that are informationally equivalent and could substitute one for the other to construct another MB and an equally-performing forecasting model. Experiments on synthetic data demonstrate that ChronoEpilogi has *near perfect average causal f1-score, with stable performance as the MTS dimensionality increases* to thousands of available TS. Moreover, the greedy approximation of the ideal ChronoEpilogi achieves a 70% speedup with only a 0.05 decrease in f1 score for causal discovery (*Claim 1*). Furthermore, ChronoEpilogi variants *are on par with Group Lasso performance* [SFHT13] (*Claim 4*) -arguably the most scalable algorithm for the single solution TVS problem-, with the added benefit of providing multiple solutions in real datasets from the Monash archive [GBW$^+$21] like Electricity and Traffic (*Claim 3*). On both real and synthetic datasets the multiple solutions produced by ChronoEpilogi have similar forecasting performance to the unique solution produced by GroupLasso when averaging across all selected targets (*Claim 5*) while when causal ground truth is

---

[1]Chronos greek and Epilogi greek
[2]https://github.com/ev07/ChronoEpilogi

available ChronoEpilogi outperforms GroupLasso in terms of causal f1-score (*Claim 2*). Finally, both variable selection algorithms actually conserve or even improve the models' performance compared to models trained on the original MTS (*Claim 6*). We also investigate how SHAP explanations of regression models might misrepresent the role of variables belonging to some but not all Markov boundaries of the modeled target. We claim that the SHAP importance of each equivalent set of variables is distributed among equivalent variables, hence leading to underestimations of the importance of equivalent sets when considered individually (Claim 7). This attribution is unstable: on different data splits, different variables among an equivalence set obtain high importance (Claim 8).

## 2  Preliminaries

We represent a multivariate time series by a set of univariate TS $\boldsymbol{X} = \{X^1, ..., X^N\}$, where each $X^i \in \mathbb{R}^M$ ($1 \le i \le N$) are regularly sampled observations of a time series variable. We denote with $X_t^i \in \mathbb{R}$ the $t^{th}$ occurrence of $X^i$, and by $X_{t-L,t-1}^i \in \mathbb{R}^L$ the last $L$ occurrences of $X^i$ before the instant $t$. Note that, in the following part, bold letters (e.g., $\boldsymbol{X}$) refer to sets of TS variables. The TS forecasting task we consider is to predict the values of a target TS variable $T_t$ ($T = X^i$ for some $i$), where $L \le t \le M$, with a model $f$ parameterized by $\Theta$ that uses up until $L$ past timestamps (for each $t$) of the multivariate TS $\boldsymbol{X}$. Hence, $T_t == f_\Theta(\boldsymbol{X}_{t-L,t-1})$.

Conditional independence of two TS variables $X_t$ and $Y_{t'}$ given a third one $Z_{t''}$, written as $X_{t'} \perp\!\!\!\perp Y_{t'} | Z_{t''}$, is defined as $P(X_t, Y_{t'} | Z_{t''}) = P(X_t | Z_{t''}).P(Y_{t'} | Z_{t''})$. In our work, we assume stationarity, namely the conditional distribution of $T_t$ as a function of $L$ previous lags of $\boldsymbol{X}$ does not depend on $t$. Stationarity implies *temporal consistency*, where the conditional independence relations do not rely on $t$. In this respect, we consider the principle of *temporal precedence* according to which causes (i.e., independent variables) occur before an effect (i.e., outcome). We exclusively test conditional independence relations between a target $T_t$ and variables with timestamps ranging between $t - L$ and $t - 1$. Due to stationarity, we can remove the index and denote $X_{t-L,t-1}$ as $X$.

**Definition 2.1** (Information Equivalence (IEQ) )**.** Two TS variables $X_a$ and $Y_b$ are information equivalent (shorthand: equivalent) with respect to a target $T_t$ given conditioning TS variable set $\boldsymbol{Z}$, noted $X_a \equiv_T Y_b | \boldsymbol{Z}$, iff they are made independent of $T_t$ by conditioning on the other and on $\boldsymbol{Z}$ itself. Formally: $X_a \equiv_T Y_b | \boldsymbol{Z} \iff X_a \perp\!\!\!\perp T_t | Y_b, \boldsymbol{Z}$ and $Y_b \perp\!\!\!\perp T_t | X_a, \boldsymbol{Z}$. We write shortly $X_a \equiv_T Y_b$ when the conditioning set is $\boldsymbol{Z} = \{\}$.

**Definition 2.2** (Markov Boundary $\mathbf{MB}(T_t)$)**.** Given a set of TS variables $\boldsymbol{X}$ and a target variable $T$ in this set, $\boldsymbol{M}(T_t)$ is a *Markov blanket* of $T_t$ iff: $T_t \perp\!\!\!\perp (\boldsymbol{V} \setminus \boldsymbol{M}) | \boldsymbol{M}$. The set $\boldsymbol{M}(T_t)$ is a *Markov boundary* of $T_t$ iff $\forall \boldsymbol{M}' \subsetneq \boldsymbol{M}, T_t \not\perp\!\!\!\perp (\boldsymbol{V} \setminus \boldsymbol{M}') | \boldsymbol{M}'$.

**Definition 2.3** (Multiple Time-series variable selection (MTVS))**.** Let $\mathbf{MB}(T_t)$ a reference MB of a target $T_t$ TS variable. The solution of the MTVS problem consists into finding all MBs, denoted by $\mathcal{M}$. All $MB(T_t)$ are information equivalent. For all $\mathbf{MB}_i(T_t) \in \mathcal{M}$ the forecasting models $T_t = f_\Theta(\mathbf{MB}_i(T_t))$ are equally-performing according to a metric (e.g., $R^2$).

**Example 2.1.** We consider a hypothetical water flow monitoring system involving two rivers $A, B$ and a small hydroelectric station on river $B$. Let MTS $X, Z \in \mathbb{R}$ be the inflows of the two confluent rivers $A$ and $B$. The dam is controlled such that if $Z_t \le z_{th}$, it does not produce energy. Otherwise, it diverts a flow $z_{th}$ to power production. The power production flow mixes with river $A$ first for total flow $Y_t = f(Z_t) + X_t$, then with the rest of river $B$ for total flow $T_t = X_t + Z_t$, with $f(Z_t) \in \{0, z_{th}\}$. In this situation, $X$ and $Y$ are deterministically related only when taking account $Z$, and information equivalent for $T$ given $Z$ only. Both $X, Z$ and $Y, Z$ are MB of $T$. In this case, solving the MTVS problem relates to discovering both subsets without misidentifying singletons and $X, Y$ as MB.

TS variables selected by a MTVS algorithm can be characterized according to whether they belong to all, at least one, or none of the Markov boundaries of a target $T_t$ [BT21]. A variable $X_a$ is said *irreplaceable* iff it is part of all information equivalent Markov boundaries of $T_t$, i.e., $X_a \in \mathbf{MB}(T_t) \forall \mathbf{MB}(T_t) \in \mathcal{M}$. A variable $X_a$ is said *replaceable* iff it is part of at least one $\mathbf{MB}(T_t)$ but not indispensible, i.e., $\exists \mathbf{MB}(T_t), \mathbf{MB}'(T_t) \in \mathcal{M}, X_a \in \mathbf{MB}(T_t) \wedge X_a \notin \mathbf{MB}'(T_t)$. A variable $X_a$ is *redundant* iff $\exists \boldsymbol{Z} \subseteq \boldsymbol{X} \setminus \{X_a\}, X_a \not\perp\!\!\!\perp T_t | \boldsymbol{Z}$ and $\forall \mathbf{MB}_i(T_t) \in \mathcal{M}, X_a \notin \mathbf{MB}_i(T_t)$ while it is called *irrelevant* iff $\forall \boldsymbol{Z} \subseteq \boldsymbol{X} \setminus \{X_a\}, X_a \perp\!\!\!\perp T_t | \boldsymbol{Z}$.

If *Causal Markov Condition* and *Faithfulness* [CMJSV$^+$23] hold, each target $T_t$ has the guarantee to have a unique Markov Boundary, which is also the unique solution of the MTVS problem. However, common probability distributions in real settings might violate faithfulness [URBY13], hence the

$\text{MB}(T_t)$ might not be unique. Jointly Gaussian random variables (RV) with a singular covariance matrix or with deterministic relations do no respect faithfulness [3] [Mee95, Lem07]. While faithfulness does not hold, the weaker *Composition* property relaxes the structure of faithful data. As a matter of fact, faithfulness implies composition, while the opposite relation does not necessarily hold [Pea88].

*Assumption* 2.1 (Composition). For any subset of RVs $\boldsymbol{X}, \boldsymbol{Y}, \boldsymbol{T}$ and conditioning set $\boldsymbol{Z}$:

$$\boldsymbol{X} \perp\!\!\!\perp \boldsymbol{T} | \boldsymbol{Z} \text{ and } \boldsymbol{Y} \perp\!\!\!\perp \boldsymbol{T} | \boldsymbol{Z} \implies \boldsymbol{X} \cup \boldsymbol{Y} \perp\!\!\!\perp \boldsymbol{T} | \boldsymbol{Z} \tag{1}$$

Composition is a general property of the joint probability distribution of a set of RVs regardless of their temporal context, hence we dropped index $t$ from formal definitions. The reciprocal property is called *Decomposition*, and it is always true in any probability distribution. Many common probability distributions that violate faithfulness actually satisfy composition.

**Example 2.2.** Consider a TS $\boldsymbol{X}$ of $n$ covariates for which composition holds. Any *deterministic transformation* of $\boldsymbol{X}$, namely $\boldsymbol{Y} = f(\boldsymbol{X})$ of size $m$, where $Y_i = f_i(X_{\sigma(i)})$, with $f_i$ invertible and $\sigma_i$ a mapping from $[|1, m|]$ to $[|1, n|]$, also satisfies composition.

**Example 2.3.** Jointly Gaussian distributions [JW07] also satisfy composition, as they are a special case where pairwise independence is equivalent to mutual independence. If independence relations $\boldsymbol{X} \perp\!\!\!\perp \boldsymbol{T} | \boldsymbol{Z}$ and $\boldsymbol{Y} \perp\!\!\!\perp \boldsymbol{T} | \boldsymbol{Z}$ hold, the union $\boldsymbol{X} \cup \boldsymbol{Y}$ contains only RVs that are pairwise independent from each variable in $\boldsymbol{T}$ given $\boldsymbol{Z}$. This implies $\boldsymbol{X} \cup \boldsymbol{Y} \perp\!\!\!\perp \boldsymbol{T} | \boldsymbol{Z}$.

For probability distributions where all information equivalences are caused by invertible deterministic transformations between individual RVs (singletons), the MBs of any target are *interchangeable*: the variables are equivalent regardless of the conditioning set. More generally, we define interchangeability for two Markov Boundaries, where each variable in a MB(T) is equivalent to a variable in the other MB(T) conditioned on the remaining MB(T) variables. We assume that all MB are interchangeable.

*Assumption* 2.2 (Interchangeability). Two MBs of a target $T_t$, $\boldsymbol{MB}(T_t)$ and $\boldsymbol{MB'}(T_t)$ are interchangeable iff:

$$\forall \boldsymbol{MB}(T_t), \boldsymbol{MB'}(T_t) \in \mathcal{M}, \forall X \in \boldsymbol{MB}(T_t), \exists Y \in \boldsymbol{MB'}(T_t), (\{X\} \cup \boldsymbol{MB'}(T_t) \setminus \{Y\}) \in \mathcal{M}$$

**Example 2.4.** In the water flow example 2.1, $Z$ is irreplaceable, and $X$ and $Y$ are replaceable by each other. The markov boundary structure $\mathcal{M} = (\{Z, X\}, \{Z, Y\})$ satisfies Interchangeability, as $\mathcal{M} = \{Z\} \times \{X, Y\}$ is a cartesian product of equivalence classes.

## 3 The ChronoEpilogi Algorithm

In this section, we present the details of our MTVS algorithm, named ChronoEpilogi. We propose two versions of the algorithm: (1) Forward Backward Equivalent (FBE - Algorithm 1 ) and (2) a computationally optimized version (approximate) named Forward Equivalent (FE - Algorithm 2).

| **Algorithm 1** ChronoEpilogi-FBE |
| --- |
| **Require:** TS $\mathbf{X}$, target $T$, max lag $L$, threshold params $[\alpha, \gamma, \delta]$ |
| 1: set $\mathbf{S} \leftarrow$ FORWARD($\mathbf{X}, T, L, \alpha$) |
| 2: $\mathbf{S} \leftarrow$ BACKWARD($\mathbf{X}, T, L, \boldsymbol{S}, \gamma$) |
| 3: set $\mathcal{M} \leftarrow$ EQUIV($\mathbf{X}, T, L, \boldsymbol{S}, \delta$) |
| 4: **return** $\mathcal{M}$ ▷ set of eq. Markov bound. |

| **Algorithm 2** ChronoEpilogi-FE |
| --- |
| **Require:** TS $\mathbf{X}$, target TS variable $T$, max lag $L$, threshold params $[\alpha, \delta]$ |
| 1: $\mathcal{M} \leftarrow$ FORW-EQUIV($\mathbf{X}, T, L, \alpha$) |
| 2: **return** $\mathcal{M}$ ▷ set of eq. Markov bound. |
| 3: |
| 4: |

In FBE, we first select informative TS random variables (RV) for predicting the values of a target $T$, adopting a greedy heuristic (FORWARD routine, line 1). Then, a backward phase iteratively removes redundant RVs from the selection (BACKWARD routine, line 2). Lastly, equivalent Markov boundaries are discovered (EQUIV routine, line 3) by checking if any of the selected RVs can be replaced by another one. In ChronoEpilogi-FE, we select equivalent Markov boundaries during informative RV selection (FORW-EQUIV routine, line 1). Such choice permits us to compute conditional independences on smaller TS variable sets. To further reduce time complexity, we also

---

[3] Conditions leading to faithfulness violations have been studied from the standpoint of probability distributions and graphs [Sad17, ZZM17] as well as applications in homeostatic systems [ZS08], evolutionary systems [And13], gene expression [SLLA13] or spatio-temporal records of meteorological phenomenons [YWD+17].

---

**Algorithm 3** FORWARD

---

**Require:** TS $\mathbf{X} \in \mathbb{R}^{N \times M}$, target $T$, max lag $L$, stopping threshold $\alpha$
1: set $\mathbf{S} \leftarrow \{T_{1,M-1}\}$
2: **model** $m \leftarrow$ fit and save $f_\Theta$ such that $T_{L+1} = f_\Theta(\boldsymbol{S}_{1,L}), ..., T_M = f_\Theta(\boldsymbol{S}_{M-L,M-1})$
3: **repeat**
4:     $R \leftarrow m.residuals \in \mathbb{R}^{M-L}$
5:     $S_{new} \leftarrow \arg\min_{X' \in \mathbf{X} \setminus \mathbf{S}}$ Lag-Pearson-pval$(R, X', L)$       $\triangleright$ Select $S_{new}$ s.t. $S_{new} \not\perp\!\!\!\perp R$
6:     $\mathbf{S} \leftarrow \mathbf{S} \cup S_{new}$
7:     **model** $m' \leftarrow m$
8:     **model** $m \leftarrow$ fit and save $f_\Theta$ on $\boldsymbol{S}$
9: **until** pvalue of Likelihood ratio test of (H0: $m' = m$) $\geq \alpha$     $\triangleright$ Verify $S_{new} \not\perp\!\!\!\perp T_{t+1} | \boldsymbol{S} \setminus S_{new}$
10: **return** $\boldsymbol{S} \setminus S_{new}$

---

**Algorithm 4** BACKWARD

---

**Require:** TS $\mathbf{X}$, target $T$, time pred. $t$, max lag $L$, sel. variables $\mathbf{S}$, stopping threshold $\gamma$
1: **repeat**
2:     **for** $S_{del} \in \boldsymbol{S}$ **do**
3:         **if** pvalue of Likelihood ratio test between $f_\Theta(\boldsymbol{S} \setminus \{S_{del}\})$ and $f_\Theta(\boldsymbol{S}) \geq \gamma$ **then**
4:             $\boldsymbol{S} \leftarrow \boldsymbol{S} \setminus \{S_{del}\}$
5: **until** no change in $\mathbf{S}$
6: **return** $\boldsymbol{S}$

---

approximate the search of equivalent Markov boundaries using forecasting model residuals. In Example 2.1, the forward phase might select an upstream redundant variable $U$ first, then $Z$ and $X$ before terminating. The backward phase would test $U \perp\!\!\!\perp T | X, Z$, removing $U$ from the selected set. Finally, the equivalence phase would test $Y \equiv_T X | Z$ and produce the solution space $\mathcal{M}$.

We evaluate ChronoEpilogi considering AutoRegressive Distributed Lags (ARDL) (linear) forecasting model [HPS84, PSS01]. An ARDL model of orders $p, q$ uses lags of both the target ($T$) and other TS RVs ($\mathbf{X}$) as predictors, with $T_t = a_0 + \sum_{i=1}^{p} a_i.T_{t-i} + \sum_{i=1}^{k} \sum_{j=1}^{q} b_j.X_{t-j}^i + \epsilon_t$. The model includes autoregressive terms ($a_j$) and other explanatory variable terms ($b_j$). It is generally required that $\epsilon_t$ is an iid centered normal noise, but this assumption can be relaxed as we can build estimators in the presence of autocorrelated noise and noise with heteroscedastic components [Whi80, HPS84]. Under sufficient assumptions, an ARDL model can be estimated using Ordinary Least Squares (OLS) procedures, and correspond to a Maximum Likelihood estimation model. Consequently, this estimation has the advantages of the OLS method, with guaranteed convergence and fast computation.

Hereafter, we detail sub-routines of ChronoEpilogi. The *forward* phase (Algorithm 3) iteratively builds a first selected set starting from the past $L$ lags of $T$ (line 1), then incrementally selects new RVs of $\mathbf{X}$. At each iteration, a selected variable ($S_{new}$) is the one maximizing the statistical importance of Pearson correlation between all windows of length $L$ (the predictors) and the residuals of model $m$ (line 5). Algo. 7 (see Appendix) contains the correlation computation pseudo-code, which iterates all the windows in a candidate variable $X'$ (line 2) to compute p-values of Pearson correlation with the forecasting model residuals. The selection in the FORWARD routine terminates when the last forecasting model (built over $\mathbf{S}$) is not statistically better than the previous one (line 9). We use Likelihood ratio test [Kin98], as they suit ARDL models. The *backward* phase (Algorithm 4) iteratively removes redundant RVs from the selected RVs in the *forward* phase. The main loop of the algorithm tests each one of these RVs. It stops as soon as removing any variable degrades the predictive performance of the *best-so-far* forecasting models (line 3), ensuring that the produced variable set is minimal and equally predictive as the *forward* phase solution. We prove that the forward and backward phases provide an exact solution of the MTVS problem (See appendix D.1,D.2). The *equivalent* search phase (Algortihm 5) tests the equivalence between each selected TS variable $\mathbf{S}$ (line 2), and any non-selected variable $S^d$ (line 4). In line 5 we compare the equivalence replacing the two Rvs in a new forecasting model. If this latter is comparable with the baseline, we store the equivalent variable sets in dictionary $Q$ (line 6).

---

**Algorithm 5** EQUIV

---

**Require:** TS $\mathbf{X}$, target $T$, time pred. $t$, max lag $L$, sel. variables $\mathbf{S}$, equivalence threshold $\delta$
1: dictionary $Q \leftarrow \{:\}$
2: **for** $S \in \boldsymbol{S}$ **do**
3:     $Q[S] \leftarrow \{\}$
4:     **for** $S^d \in (\mathbf{X} \setminus \mathbf{S})$ **do**
5:         **if** pvalue of LR test between $f_\Theta(\{S^d\} \cup \boldsymbol{S} \setminus \{S\})$ and $f_\Theta(\{S^d\} \cup \boldsymbol{S}) \geq \delta$ **then**
6:             $Q[S] \leftarrow Q[S] \cup \{S^d\}$
7: $\mathcal{M} \leftarrow \{S_1\} \cup Q[S_1] \times ... \times \{S_n\} \cup Q[S_n]$
8: **return** $\mathcal{M}$

---

---

**Algorithm 6** FORW_EQUIV

---

**Require:** MTS $\mathbf{X}$, target $T$, maximal lag $L$, stopping threshold $\alpha$, equivalence threshold $\delta$
1: set $\mathbf{S} \leftarrow \{T_{1,M-1}\}$
2: set $\boldsymbol{R} \leftarrow \boldsymbol{X} \setminus \boldsymbol{S}$
3: dictionary $Q \leftarrow \{:\}$
4: **model** $m \leftarrow$ fit and save $f_\Theta$ such that $T_{L+1} = f_\Theta(\boldsymbol{S}_{1,L}), ..., T_M = f_\Theta(\boldsymbol{S}_{M-L,M-1})$
5: **repeat**
6:     $Res \leftarrow m.residuals \in \mathbb{R}^{M-L}$
7:     $S_{new} \leftarrow \arg\min_{\{X' \in \boldsymbol{R}\}}$ Lag-Pearson-pval$(Res, X', L)$    ▷ Select $S_{new}$ s.t. $S_{new} \not\perp Res$
8:     $\boldsymbol{S} \leftarrow \boldsymbol{S} \cup \{S_{new}\}$
9:     $\boldsymbol{R} \leftarrow \boldsymbol{R} \setminus \{S_{new}\}$
10:     $Q[S_{new}] \leftarrow$ FIND-EQUIVALENCES$(Res, S_{new}, \boldsymbol{R}, \delta)$   ▷ Test $C \equiv_{Res} S_{new}$ for $C \in \boldsymbol{R}$
11:     $\boldsymbol{R} \leftarrow \boldsymbol{R} \setminus Q[S_{new}]$
12:     $m' \leftarrow m$
13:     **model** $m \leftarrow$ fit and save $f_\Theta$ on $\boldsymbol{S}$
14: **until** pvalue of Likelihood ratio test of (H0: $m' = m$) $\geq \alpha$    ▷ Verify $S_{new} \not\perp T_{t+1}|\boldsymbol{S} \setminus S_{new}$
15: **return** $\boldsymbol{S} \setminus S_{new}, Q$

---

Once the equivalent Markov Boundaries $\mathcal{M}$ are obtained, we can characterize irreplaceable variables and replaceable variables, as irreplaceable variables are the unique members of their equivalence class. Conversely, non-unique variables in their equivalence class are replaceable. In Section D (see Appendix), we provide soundness and completeness proofs of the multiple solutions computed by Algortihm 5. In Algorithm 6 we report the pseudocode of FORW_EQUIV routine, which is used in FE to select informative variables and to estimate equivalent sets over forecasting model residuals. Algorithm 8 in Appendix details the residual-based equivalence search (FIND-EQUIVALENCES) employed during the forward phase (Algorithm 6, line 10). Such a routine tests statistical redundancy by modeling residuals with model $g_\Theta(.)$. In this sense, if a non-selected variable $X$ is equivalent to the tested variable, $S_{new}$ is added to the equivalent set (line 7).

**Complexity analysis** Given a TS $\mathbf{X} \in \mathbb{R}^{N \times M}$, and denoted by $\mathbf{S}$ the selected TS variables in the forward phase, both ChronoEpilogi versions ($FBE$ and $FE$) run $\mathcal{O}(N|\mathbf{S}|)$ conditional tests. In the FORWARD phase, the sub-routine Lag-Pearson-pval (Algorithm 7) runs in $\mathcal{O}(L \, n \, log(n))$ if Fast Fourier Transform is adopted [MZZ$^+$22], where $L$ is the number of lags to predict a target and $n = M - L$ the size of residuals (number of predicted targets). Fitting an ARDL model through matrix inversion requires $\mathcal{O}(|S|^3 \, L^3 \, M)$. Overall, the forward, backward, equivalent phases take $\mathcal{O}(|S| \, N \, L \, M + |S|^4 \, L^3 \, M)$, $\mathcal{O}(|S|^5 \, L^3 \, M)$, $\mathcal{O}(|S|^4 \, L^3 \, N \, M)$ time respectively.

The two variants of ChronoEpilogi, satisfy different properties. FBE (Alg. 1) is an ideal version of our algorithm with provable soundness and completeness under mild assumptions. FE (Alg. 2) is a greedy approximation of FBE. To simplify the proofs of FBE theoretical guarantees we rely on a practical commonly made in causal discovery algorithms that the conditional independence tests are correct [4] [PNBT07, BT21]. We should stress this requires considering different statistical tests for different distributions [SP20]. We prove that FBE is **sound** as any set of TS variables returned as a solution in $\mathcal{M}$ is a Markov blanket. In other words, our algorithm does not return false positives. We

---

[4]Alternatively, algorithms might rely on an oracle producing variable sets with given properties [SLLA13].

then examine the conditions under which FBE is **complete**, i.e. all Markov boundaries are solutions in $\mathcal{M}$ discovered by the algorithm. This corresponds to the lack of false negatives in the equivalent Markov boundary discovery. Theorems proofs are given in Appendix D.

**Theorem 3.1** (Soundness and Completeness of FBE). *Assuming composition, interchangeable Markov boundaries, and perfect tests for the correlation, termination, elimination and equivalences, FBE computes all Markov boundaries $\mathcal{M}$ of a target $T_t$ and only Markov boundaries.*

The BACKWARD, EQUIV and FORWARD phases rely on model-based independence tests that might be replaced by any conditional independence (CI) test. In other words, CI tests performed in line 5 of the FORWARD algorithm (Algorithm 3) compute correlations over forecasting model residuals. We argue that in the context of linear regression with joint normal variables, measuring the residual correlation is equivalent to a model-based conditional independence test. This is the case of linear models of joint normal variables for which $R \perp\!\!\!\perp Z$, and $R$ is normal jointly with all other variables. We notice that residual-based tests are not necessarily equivalent to full conditional tests. Yet no conditional independence remains undetected by a residual test. Formally:

**Theorem 3.2.** *Given $T$ and $X$, two TS variables and $Z$ a conditioning set. Let $R$ be the residual variable corresponding to the regression of $T$ on $Z$. Assume that the modeling achieves independence of the residuals: $R \perp\!\!\!\perp Z$, then: $T \perp\!\!\!\perp X | Z \implies R \perp\!\!\!\perp X$. Additionally, if for all candidate variable $X$, $R \perp\!\!\!\perp X \implies R \perp\!\!\!\perp X, Z$, then $R \perp\!\!\!\perp X \implies T \perp\!\!\!\perp X | Z$.*

## 4    Experimental Framework

**Synthetic dataset**: To build synthetic datasets with multiple Markov boundaries that respect composition and interchangeability, we build MTS starting from faithful distributions. Each MTS starts from a Vector AutoRegressive process with 20 variables, with different maximal lags and Markov boundary sizes. We then make copies of some of the MB variables to obtain replaceable variables. Those copies can be randomly shifted forward to change the lag of the relation. The same process is used to obtain redundant variables, copying correlated features not in the original Markov boundary. Irrelevant variables are then added to the dataset by sampling other VAR processes. For each MB size [2, 5, 10], total number of variables [10,100,1000], and maximal lag [1,5,10], we sample 10 datasets per configuration. We verify that the noise intensity varies, as the $R^2$ of a model trained with a MB ranges from 0.02 to 0.9 over the different data instances.

**Real datasets** We evaluate our approach on five forecasting datasets covering different domains: Electricity (consumption), Solar (production), S.F. Traffic [GBW$^+$21], METR-LA, and PEMS-BAY [LYSL18] (transport). They are commonly used in recently proposed deep forecasting models [JZL$^+$23, ZY22, WCWW22]. Electricity has 321 TS and 26304 observations, Traffic 862 TS and 17544 observations, Solar 137 TS and 52560 observations, METR-LA 207 TS and 34272 observations, PEMS-BAY 325 TS and 52116 observations. We evaluate 10 randomly chosen targets per dataset.

**Cross Validation protocol** In TS data, observations are generally not independent, so data splitting for forecasting tasks must ensure that the train split precedes the test split (Forward Chaining Cross Validation). Therefore, we split the dataset along time into a Tuning and Holdout set, respectively for training and evaluating feature selection algorithms and forecasting models. The Tuning set is itself separated into five folds along the time axis, to conduct hyperparameter optimization. We optimize each considered pair of TS selection algorithm and forecasting algorithm together, for maximal average predictive performance $R^2$ over all folds. Due to the consequent training time of deep learning models, it is impractical to tune TSS (Time Series variables Selection) algorithms together. We first tune each TSS algorithm with Support Vector Regression (SVR) models as proxies, then tune deep forecasters for the tuned TSS parameters.

**Baselines and Forecasting models** We compare our algorithm with the only linear scalable baseline, GroupLasso [NST13], and with no selection in the case of Electricity, Traffic, and Solar. For forecasting, we use the ARDL model [PSS01], as it is a standard linear model for MTS data. Real datasets are forecasted using nonlinear models: Support Vector Regression (SVR) with nonlinear kernel, and deep forecasters like DeepAR [SFGJ20] and Temporal Fusion Transformer (TFT) [LALP21]. The input window size is 10 for synthetic MTS and 96 for the five real datasets, similarly to a recent benchmark [SWX$^+$23]. The tuned parameters are described in Appendix (Table 3). We report the performance of the best forecasting model for each target.

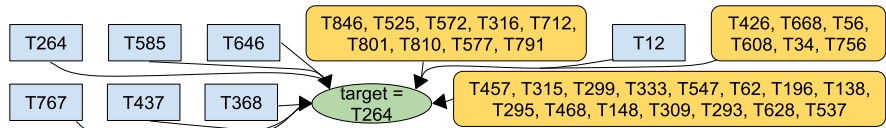

Figure 1: Multiple solutions of ChronoEpilogi on target T264 of Traffic. MB are build by combining irreplaceable variables (in blue) with members of the replaceable equivalence class (in yellow).

**Evaluation metrics** We evaluate the quality of Markov boundaries based on three criteria: computation time, predictive performance, size, and on synthetic datasets, f1-score using causal ground truth. Predictive performance is measured by $R^2$, $RMSE$ and $MAPE$. When more than 20 multiple solutions are obtained by ChronoEpilogi, we sample 20 solutions among the solution space, evaluate each of them, and report the mean. When the causal graph is known, we measure the f1-score of identifying respectively irreplaceable and replaceable TS variables. Since GroupLasso returns a unique solution, we also compute the f1-score of identifying all causal variables (irreplaceable and replaceable). Formally, denoting TP the number of correctly detected variables, FN for undetected correct variables, FP for wrongly detected variables, $f1 = \frac{2TP}{2TP+FP+FN}$.

**SHAP explanations** To assess the impact of replaceable variables on SHAP explanations, we train Support Vector Regressors (SVR) and XGBoost Regressors (XGB) for each of the 90 synthetic MTS with 100 covariates (denoted as *full MTS*), using a window of 10 timesteps. We then compute the SHAP values of random holdout samples using the KernelSHAP approximation [LL17], which we sum along the time axis to obtain the overall importance of variables (per additivity property of SHAP values [KVSF20]). We consider the average of those values as global explanations [BSC+21]. We obtain average SHAP values $s_1^i...s_{100}^i$ corresponding to the global importance of each variable in the MTS number $i$. The total importance of each equivalent variable set $S$ becomes $s_{tot} = \sum_{i \in S} |s_i|$. Then, we compute the contribution percentage of the most important variables (until top-14) in the equivalent set $|s_i/s_{tot}|$. For comparison with SHAP explanations over a unique MB, we create *reduced MTS* where we remove all but one replaceable variable per equivalent set. For instance, if in MTS $i$ variables $\{1, 4, 16, 78\}$ are equivalent, we remove $\{4, 16, 78\}$ and compute the reduced importance $sr_1^i$ of the representative 1. We obtain instability results by repeating model training and explanation over different (time-contiguous) data splits of each MTS. Stability is measured as proposed by Nogeira et al. [NSB18], applied to the top $|MB|$ variables of each explanation, with $|MB|$ the theoretical size of a Markov Boundary. The metric accounts for covariate size, hence results are comparable between full MTS and reduced MTS.

## 5   Experimental Results

Variant FBE has near perfect average causal f1-score (Table 1), with stable performance as the number of TS increases 2b). **Claim 1**: Variant FE has a 70% speedup compared to FBE with only a decrease of causal f1 metric of 0.05 (Table 1), due to a less precise equivalence discovery phase leading to lower replaceable f1-score (Fig.2b). As the number of TS grows, FE runtime grows slower than FBE (Fig.2a). Given its high comparative effectiveness and efficiency, we decided to use FE on real datasets. **Claim 2**: FBE and FE achieve better results than GroupLasso (GL) w.r.t. causal f1-score, despite similar $R^2$ scores (Table 1). We attribute this difference to GL sensitivity in tuning regularization parameters, compared to ChronoEpilogi thresholds (see Appendix Fig.5). **Claim 3**: FE uncovers multiple MBs in Electricity (for 3 targets) and Traffic (for 7 targets), PEMS-BAY (3 targets), METR-LA (7 targets). In Solar only one MB is found for all targets. The average size of selected TS variables (Table 2) is less than 4% of each MTS. The maximal selected size is below 5% all except Electricity (11%). The solutions have close to identical $R^2$ in Electricity, and spread over 0.02 around the median for Traffic (Fig.2c). Fig.1 illustrates a compact representation of the solution space for one target of Traffic. **Claim 4**: On averaging across all targets the multiple solutions produced by ChronoEpilogi-FE have similar predictive performance to the unique solution produced by GroupLasso (GL). We use the total 50 targets to conduct a Friedman-Nemenyi statistical test on method ranking [Her20], and conclude that GL and FE are not statistically different, but **Claim 6**: variable selection do improves models' performance compared to models trained on the original MTS (full analysis Appendix E.2). **Claim 5**: In Electricity and Solar, GL produces slightly smaller solutions than FE while in the rest datasets, FE produces 8 times smaller solutions than GL (Table 2).

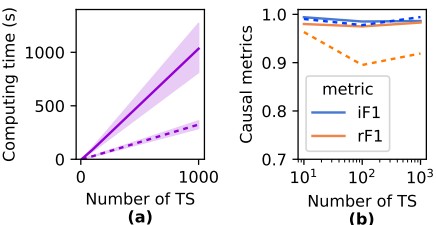
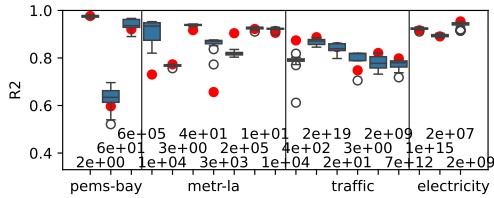

(a) computing time (b) f1 score for replaceable (rF1) and irreplaceable (iF1) variables of FBE (solid lines) and FE (dashed lines) over the synthetic dataset. FE has up to 70% speedup while losing at most 0.1 in rF1.

(c) Boxplot of the $R^2$ of the 20 sampled MB for each target where ChronoEpilogi finds multiple MBs. Red dots figure the performance of GroupLasso. The number of MB detected is reported for each target.

Figure 2: Performance of ChronoEpilogi versions on (a)(b) synthetic MTS and (c) real MTS.

Table 1: Computation, predictive and causal performance of tuned ChronoEpilogi variants (FBE, FE) and GroupLasso (GL) on the synthetic dataset, over the 270 synthetic MTS. FE has comparable execution time and predictive power to GL, with a 30% increase in causal f1-score.

|  | Time(s) | $R^2$ | MB size | Number of MB | causal f1-score |
|---|---|---|---|---|---|
| FBE | 375±772 | 0.373±0.236 | 5.69±3.33 | 8.87e+05±3.7e+06 | 0.99± 0.064 |
| FE | 118±178 | 0.373±0.236 | 5.74±3.31 | 6.87e+07±7.2e+08 | 0.94± 0.188 |
| GL | 120±149 | 0.337±0.239 | 7.50±10.2 | NA | 0.57± 0.349 |

Note that computing one solution with FE forward phase is up to two orders of magnitude faster than GL. Table 2 reports the best forecaster after careful hyperparameter tuning per target.

**Shapley values instability over replaceable variables** **Claim 7**: Models tend to select one or a few variables as important predictors. For XGB models, we report in Fig.9 (see Appendix) the average contribution of the top important variables for each equivalent set of variables. Clearly, **importance is increasingly shared as the number of equivalent variables grows. Claim 8**: The most important variable of an equivalent set is highly unstable to data resampling. The stability [NSB18] of the top ranked variable is respectively 0.05±0.11 and 0.02±0.09 for XGB and SVR, where -0.05 means total randomness and 1 deterministic selection. Additionally, the presence of replaceable variables impacts the stability of the important variables of the entire explanation. The top $|MB(T)|$ ranked variables overall over *full MTS* are significantly more unstable compared to *reduced MTS*. We applied paired Wilcoxon signed rank test and concluded that top variables in full MTS are less stable than when a unique MB(T) is left, with p-value 4.2e-13.

## 6 Related work

Most of the state-of-the-art variable selection algorithms for MTS have focused on selecting a unique concise solution. Scalable methods include using bivariate linear VAR models to select causally related pairs of TS, without considering multivariate interactions [SLL$^+$15]. There are mRMR approaches for MTS [HRL15] with Dynamic Time Wrapping distances [RGFO17], but with a quadratic complexity in the number of TS due to distance computations, which are proven to be impractical for large dataset sizes. We selected GroupLasso [NST13] as baseline in our experiments, as it can specify groups of TS [LNZ15][HMNK10] while applying Lasso-type optimization with linear modeling. It is worth noticing that Lasso-type algorithms have been extended to identify multiple solutions in i.i.d data [PLCT18] without, however, proving any formal property.

As a matter of fact, the problem of computing multiple solutions for the variable selection problem is still in its infancy. In particular, we are not aware of any algorithm applied to MTS data. Theoretically sound methods focus on different distributional properties and assumptions. TIE* [SLLA13] assumes that a single Markov boundary discovery algorithm for *non-faithful* data can be called as an oracle. All $MB(T)$ are discovered using exhaustive combinatorial conditional tests ($\mathcal{O}(s.n^s)$ where $s$ is the maximal size of a considered MB and $n$ the number of variables [BT21]). TMFBS [BT21] proposes forward-backward MB discovery algorithms that aims to decrease the number of redundant

Table 2: Forecasting performance of ChronoEpilogi (FE), GroupLasso (GL) and No Selection (NS) over real datasets. We report the number of times each model was selected (TFT/DeepAR/SVR), and the time spent in the forward/equivalence phases (time F/E). FE multiple solutions are on par to the performance of the unique GL solution while their size in Traffic is 8 time smaller than GL.

| Dataset | TSS | $R^2 \uparrow$ | rmse $\downarrow$ | mape $\downarrow$ | size | time F/E | #model | #MB |
|---|---|---|---|---|---|---|---|---|
| Electricity | FE | **0.940** | **0.226** | **2.388** | 10.9 | **31** /2411 | 6/0/4 | 1e+14 |
| Electricity | GL | 0.934 | 0.236 | 2.649 | **6.5** | 127 | 3/0/7 | NA |
| Electricity | NS | 0.863 | 0.349 | 4.935 | 321.0 | NA | 0/0/10 | NA |
| Solar | FE | **0.985** | **0.109** | 0.664 | 5.1 | **3** /261 | 0/0/10 | 1 |
| Solar | GL | **0.984** | **0.111** | 0.623 | 4.7 | 119 | 0/0/10 | NA |
| Solar | NS | 0.968 | 0.159 | 1.607 | 138.0 | NA | 0/0/10 | NA |
| Traffic | FE | 0.783 | 0.442 | 39.745 | **11.7** | **12** / 1248 | 6/0/4 | 7e+24 |
| Traffic | GL | **0.797** | **0.431** | **28.130** | 79.1 | 211 | 3/0/7 | NA |
| Traffic | NS | 0.740 | 0.491 | 42.371 | 863.0 | NA | 1/0/9 | NA |
| PEMS-BAY | FE | 0.860 | 0.358 | **1.208** | **4.1** | **143**/6250 | 8/0/2 | 6e+4 |
| PEMS-BAY | GL | **0.867** | **0.355** | 1.217 | 37.7 | 765 | 10/0/0 | NA |
| PEMS-BAY | NS | 0.820 | 0.957 | 1.343 | 325 | NA | 9/0/1 | NA |
| METR-LA | FE | 0.886 | 0.374 | 1.121 | **3.8** | **100**/2246 | 8/0/2 | 2e+4 |
| METR-LA | GL | 0.864 | 0.401 | 1.230 | 41.9 | 306 | 7/0/3 | NA |
| METR-LA | NS | **0.896** | **0.363** | **1.054** | 207 | NA | 10/0/0 | NA |

conditional independence tests operated by TIE* to $\mathcal{O}(n^s)$. KIAMB is an iterative forward-backward algorithm [PNBT07] that requires *composition* and heuristic selections to find one MB of a target, with every MB having a non-zero probability of selection. Identifying all MBs might require an exponential number of runs of KIAMB [SLLA13]. In construct, ChronoEpilogi computes provable equivalent subsets of variables in $\mathcal{O}(ns)$ conditional tests under composition and interchangeability assumptions, while the forward-backward phase is based on a heuristic. We should mention that there also exist related works that rely on pure associational criteria where heuristics are used without any causal guarantee, especially in the field of gene expression [LLZ10a] [LLZ10b] (see [SLLA13] for an in-depth review). Finally, several works highlight the importance of a structured representation of multiple MBs for interpretability concerns, especially in the presence of an exponential number of MBs of a target of interest [TLT18][LPL+23]. ChronoEpilogi is the first algorithm that provides a compact representation of mutually equivalent variables for MTS. Redundant and irreplaceable variables can be easily distinguished at first glance (see Figure 1).

## 7 Conclusions

Overall, the paper's contributions are (a) the adaptation to time series and further development of the theory of multiple variable selection, (b) the design of the first variable selection algorithm for time-series data, called ChronoEpilogi, that scales to thousands of available series, (c) the conditions of soundness and completeness of ChronoEpilogi, and (d) the empirical evaluations of ChronoEpilogi demonstrating the presence of multiple solutions in real data, achieving on par performance against Group Lasso and no selection, while reducing the number of TS required to build the model by 96% (on average) by conserving or even improving forecasting performance. The reduced model could be employed as the final model for production, or as a surrogate to a model using all available time series (assuming it is better performing) to facilitate interpretation, visualization, and explanations. Finally, we leave as future work the construction of ensemble models trained on several or all MBs as a means to create forecasters more robust to noise, faulty sensors, or other systematic errors.

We note that the conclusions are limited to the scope of the experimental study. The latter could benefit from a larger scope of experiments with more real and synthetic datasets of varying size, statistical properties, and data types (e.g., discrete time-series). In addition, different variants of the main algorithm could employ non-linear models to compute residuals and non-linear correlation methods for heuristically selecting time series. The implementation has not been optimized at the low level to reach higher computational gains. Other distributional properties that could lead to greedy yet sound and complete multiple time series selection algorithms could be explored. We also hope to relax the assumption of stationarity in future works, as most practical applications have trends/seasonality, different train-test distributions, or change points.

## Acknowledgements

This work has received funding from the Horizon Europe Framework Programme under Grant agreement No 101135775 (PANDORA) and ANR under the Grant agreement 24-CE23-6509 (AIDA).

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

# A   Additional information on preliminaries

**Information Equivalence**   We remark that the equivalence between variables properly defines an equivalence relation. From its definition, it is straightforwardly reflexive ($X \equiv_T X|\boldsymbol{Z}$), symmetric ($X \equiv_T Y|\boldsymbol{Z} \implies Y \equiv_T X|\boldsymbol{Z}$) and transitive.

## A.1   Properties

Independence relationships validate certain properties in any distribution. Contraction and weak union [Pea88] link independences of sets of variables to conditional independences:

**Lemma A.1** (Weak Union). *Let $\boldsymbol{X}, \boldsymbol{Y}, \boldsymbol{Z}, \boldsymbol{W}$ be sets of variables, then $\boldsymbol{X} \perp\!\!\!\perp \boldsymbol{Y} \cup \boldsymbol{W}|\boldsymbol{Z} \implies \boldsymbol{X} \perp\!\!\!\perp \boldsymbol{Y}|\boldsymbol{Z} \cup \boldsymbol{W}$.*

**Lemma A.2** (Contraction). *Let $\boldsymbol{X}, \boldsymbol{Y}, \boldsymbol{Z}, \boldsymbol{W}$ be sets of variables, then $\boldsymbol{X} \perp\!\!\!\perp \boldsymbol{Y}|\boldsymbol{Z}$ and $\boldsymbol{X} \perp\!\!\!\perp \boldsymbol{W}|\boldsymbol{Z} \cup \boldsymbol{Y} \implies \boldsymbol{X} \perp\!\!\!\perp \boldsymbol{Y} \cup \boldsymbol{W}|\boldsymbol{Z}$.*

Equivalent variables conditioned on a certain set have the same conditional information on the target. They also contain the same information on the target when considering only information that they have in common with another variable $B$. Precisely, the following property holds for any probability distribution:

**Lemma A.3** (Lemiere [Lem07]). *Let $X, Y, T, B$ be random variables and $\boldsymbol{Z}$ a conditioning set. If $X \equiv_T Y|\boldsymbol{Z}$, then:*

$$B \perp\!\!\!\perp T|\boldsymbol{Z}, X \iff B \perp\!\!\!\perp T|\boldsymbol{Z}, Y \tag{2}$$

Joint normal distributions have a specific independence structure: pairwise independence implies mutual independence in such distributions, which is not the case generally.

**Lemma A.4.** *[JW07] Let $\boldsymbol{U}, \boldsymbol{V}$ be two sets of random variables such that $\boldsymbol{U} \cup \boldsymbol{V}$ is joint normal. Then:*

$$\boldsymbol{U} \perp\!\!\!\perp \boldsymbol{V} \iff \forall U \in \boldsymbol{U}, \forall V \in \boldsymbol{V}, U \perp\!\!\!\perp V$$

When the covariance matrix of a joint gaussian distribution is invertible, the distribution is faithful (factorising into a bayesian network). In this circumstance, there is a unique Markov boundary. For equivalence relations to exist in jointly gaussian distributions, the covariance matrix must be singular.

## A.2   Assumptions

**Composition**   Equivalent search algorithms [SLLA13], [PNBT07] can alternatively rely on a local version of composition holding only for the target variable.

*Assumption* A.1 (Local Composition [SLLA13]). The local composition assumption with respect to a target $T$ holds when for all $\boldsymbol{X}, \boldsymbol{Y}, \boldsymbol{Z}$, when Eq.1 holds with $\boldsymbol{T} = T$.

**Limitations of Composition**   Composition is essentially a weaker version of "pairwise independence implies mutual independence". Mutual independence would require that the joint probability $\mathbb{P}[\boldsymbol{X}, \boldsymbol{Y}, \boldsymbol{T}]$ can be factored into $\mathbb{P}[\boldsymbol{X}]\mathbb{P}[\boldsymbol{Y}]\mathbb{P}[\boldsymbol{T}]$ when $\boldsymbol{X}, \boldsymbol{Y}, \boldsymbol{T}$ are mutually independent. Instead, composition requires that when $\boldsymbol{X}, \boldsymbol{T}$ and $\boldsymbol{Y}, \boldsymbol{T}$ are pairwise independent, the joint probability $\mathbb{P}[\boldsymbol{X}, \boldsymbol{Y}, \boldsymbol{T}]$ factorizes as $\mathbb{P}[\boldsymbol{X}, \boldsymbol{Y}]\mathbb{P}[\boldsymbol{T}]$.

The typical example of a XOR operation on two Bernouilli variables $X, Y$ with probability $0.5$, and $T = X \mathrm{xor} Y$ invalidates composition. Indeed, $X$ and $Y$ individually bring no information on $T$, while together define entirely $T$.

Another way of seeing composition is regarding interaction among variables. In clinical trials, it is often the case that the causal effect of two context variables $X, Y$ is greater (or lesser) than the sum of the individual effects of $X$ and $Y$. This is called interaction. Composition asks that two variables $X$ and $Y$ that are individually independent from $T$, cannot interact such that $T$ becomes dependent of them. Note that interaction is possible in the case of any of $X$ and $Y$ already being dependent on $T$.

**Interchangeability** Interchangeability asks for the set of multiple solution $\mathcal{M}$ to factorize as a cartesian product of equivalence class of individual variables. This allows the solution space to be quickly discovered by a linear time algorithm.

In general, equivalences can occur at the level of group of variables, and not just individual variables. For instance, let $Z$ be a standard normal random variable. Define $X = \mathbb{I}[Z > 0].Z$ and $Y = \mathbb{I}[Z < 0].Z$. Since $X$ has the value of $Z$ if $Z > 0$ and $Y$ the converse, then $Z = X + Y$. In this situation, for any target, $Z \equiv_T X, Y$. Interchangeability requires that $Z$ cannot be part of a markov boundary of $T$.

It is possible to relax Interchangeability to a looser version by introducing additional complexity to the algorithm. If instead of testing singletons for equivalences, the algorithm also tests pairs of variables, more complex equivalence relations can be integrated. To take into account all possible equivalence relations among group of variables, exponential search is required [SLLA13].

## B  Pseudocode of Lag-Pearson-pval routine

Algorithm 7 details the Pearson correlation based test. We note that in this version, a p-value is obtained for each of the lags, the minimum of which is returned. This procedure returns p-values that are likely lower than the actual p-value, due to using the minimum as an estimator. Hence, testing for previous and current model statistical difference is kept separate from the correlation routine, ensuring that Forward phase termination relies on a single p-value.

---

**Algorithm 7** Lag-Pearson-pval

---

**Require:** residuals $R$, candidate variable $X' \in \mathbb{R}^M$, maximum lag $L$
1: $p \leftarrow 1$
2: **for** $1 \leq l \leq L$ **do**
3:     $p \leftarrow \min(p, \text{Pearson-corr}(R, X'_{l,M-L+l-1}))$
    **return** $p$

---

## C  Pseudocode of equivalence computation of version FE

The equivalence search (Algorithm 8) of the approximate FE version builds surrogate models $g_\Theta$ to predict the residuals instead of the target $T$. Its structure correspond to the definition of Information Equivalence, in that it tests proxies for the two assertions $X \perp\!\!\!\perp Res | S_{new}$, and $S_{new} \perp\!\!\!\perp Res | X$. Establishing equivalence for residuals is not always guaranteed to ascertain equivalence for $T$ (see Theorems), but we empirically find it a good proxy for our synthetic datasets.

---

**Algorithm 8** FIND-EQUIVALENCES

---

**Require:** residuals $Res$, selected variable $S_{new}$, remaining variables $\boldsymbol{R}$, maximum lag $L$, equivalence threshold $\delta$
1: set $\boldsymbol{E} \leftarrow \{\}$
2: **for** $X \in \boldsymbol{R}$ **do**
3:     $p \leftarrow$ pvalue of Likelihood ratio test between $Res = g_\Theta(X \cup S_{new})$ and $Res = g_\Theta(S_{new})$
4:     **if** p $\geq \delta$ **then**                              ▷ $X$ is redundant given $S_{new}$ to predict $Res$
5:         $p \leftarrow$ pvalue of Likelihood ratio test between $Res = g_\Theta(X \cup S_{new})$ and $Res = g_\Theta(X)$
6:         **if** p $\geq \delta$ **then**                          ▷ $S_{new}$ is redundant given $X$ to predict $Res$
7:             $\boldsymbol{E} \leftarrow \boldsymbol{E} \cup \{X\}$
    **return** $\boldsymbol{E}$

---

## D  Theorems and proofs

**Theorem D.1** (Forward Soundness). *Assuming composition (Assumption 2.1) and perfect statistical tests for the correlation and termination, the set $\boldsymbol{S}$ returned by the FORWARD and by the FORWARD-EQUIVALENCES procedures is a Markov blanket of $T_t$.*

*Forward Soundness.* We reuse the proof of KIAMB [PNBT07]. At each iteration, the association heuristic selects $S_{new}$ such that $S_{new} \not\perp\!\!\!\perp T_t | S$ whenever such a variable exists. The learning model used will improve in this case and only in this case. When the forward phase ends, for all remaining variables $C \in R$, then $C \perp\!\!\!\perp T_t | S$. By composition, this ensures that $R \perp\!\!\!\perp T_t | S$, hence $S$ is a Markov Blanket of the target $T_t$.

$\square$

**Theorem D.2** (Backward Soundness). *Assuming composition (Assumption 2.1) and perfect tests for the correlation, termination, and elimination, the set $S$ returned by the succession of the FORWARD and BACKWARD procedures is a Markov boundary of $T_t$.*

*Backward Soundness.* We reuse the proof of KIAMB [PNBT07]. We assume that the set $S$ produced by the forward phase is a Markov Blanket of $T_t$. Assuming that every conditional independence test of $T_t \perp\!\!\!\perp S_{del} | S \setminus \{S_{del}\}$ is correct. At each deletion step, assuming $V \setminus S \perp\!\!\!\perp T_t | S$ and $T_t \perp\!\!\!\perp S_{del} | S \setminus \{S_{del}\}$. In this case, by contraction (A.2), $T_t \perp\!\!\!\perp S_{del} \cup (V \setminus S) | S \setminus \{S_{del}\}$. This guarantees that $S \setminus \{S_{del}\}$ is still a Markov blanket.

By the end of the backward phase, for all $S \in S$, $T_t \not\perp\!\!\!\perp S | S \setminus \{S\}$. By contradiction, suppose that there is $M \subset S$ a Markov blanket of $T_t$. Let $X \in S \setminus M$, let $Y = S \setminus M \setminus \{X\}$. Since $M$ is a Markov blanket of $T_t$, $X \perp\!\!\!\perp T_t | M$ and $Y \perp\!\!\!\perp T_t | M$. Applying composition, $X, Y \perp\!\!\!\perp T_t | M$, then applying weak union (A.1), $X \perp\!\!\!\perp T_t | M \cup Y$. This contradicts the end condition of the backward phase, so the $S$ has to be minimal for inclusion among Markov blankets. $\square$

**Theorem D.3** (Soundness of FBE). *Assuming composition and perfect tests for the correlation, termination, elimination and equivalences , any $MB_1...MB_n$ in the set $\mathcal{M}$ computed by FBE is a Markov blanket of $T_t$.*

*FBE Soundness (Thm.D.3).* In the following, we define a useful notation $S_k$ where $S$ is an indexed set from 1 to $n$ and $1 \le k \le n$, as the set of elements of $S$ indexed from 1 to $k$.

First, we remark that it is enough to prove that $MB_n \equiv_T S_n$, where $MB_n$ is one of the produced solutions $MB_1, ..., MB_n \in \mathcal{M}$. In that case, since $S_n$ is a Markov blanket, for any variable of the MTS $B \in X$, then $B \perp\!\!\!\perp T_t | S_n$ holds. By Lemma A.3 applied to the equivalence $MB_n \equiv_T X_n$, we have $B \perp\!\!\!\perp T_t | MB_n$. Hence, $MB_n$ is a Markov blanket of $T_t$.

To prove $MB_n \equiv_T S_n$, we proceed by recursion. Suppose that $MB_{k-1} \equiv_T S_{k-1} | S_k...S_n$. We seek to prove that $MB_k \equiv_T S_k | S_{k+1}...S_n$. For simplicity of notation, in the following proof, denote $Z = S_{k+1}...S_n$. By definition of equivalence, we need to prove:

$$\begin{cases} MB_k \perp\!\!\!\perp T_t | S_k, Z \\ S_k \perp\!\!\!\perp T_t | MB_k, Z \end{cases}$$

The first condition is true since $S_n$ is a Markov Blanket of $T_t$. To prove the second condition, by the property of decomposition, and by the assumption of composition (Assumption 2.1), it is necessary and sufficient to prove that

$$S_k \perp\!\!\!\perp T_t | MB_k, Z \tag{3}$$
$$S_{k-1} \perp\!\!\!\perp T_t | MB_k, Z \tag{4}$$

By using Lemma A.3 with equivalence $MB_{k-1} \equiv_T S_{k-1} | S_k, Z$ (recursion supposition), equation (3) is equivalent to $S_k \perp\!\!\!\perp T_t | S_n^{-k} \cup \{MB_k\}$ which is the condition to include $MB_k$ in the equivalence of $S_k$ in our algorithm.

By using Lemma A.3 with equivalence $MB_k \equiv_T S_k | S_n^{-k}$ (implied by the test in the algorithm), equation (4) is equivalent to $S_{k-1} \perp\!\!\!\perp T_t | MB_{k-1}, S_k, Z$. This last relation is true due to the recursion supposition.

The initial condition holds since $S_1 \equiv_T MB_1 | S_2...S_n$. Hence, $MB_n \equiv_T MB_n$.

$\square$

**Corollary D.3.1** (of Thm.D.3). *Assuming composition, assuming that all Markov boundaries have the same size and perfect tests for the correlation, termination, elimination and equivalences, any equivalent solution is a Markov boundary.*

*Corollary.* All returned solutions are Markov Blankets (Thm.D.3), and at least one is a Markov Boundary (Thm.D.2). Since by definition of Markov Boundaries, any Markov Blanket is the superset of at least one Markov Boundary, then all returned solutions must be of the same size as the Markov Boundary contained. Hence, all solutions are Markov Boundaries. □

*Completeness (Thm.3.1).* Direction $\implies$ : suppose that the equivalence search starting from a selected set $\boldsymbol{S}$ find out all Markov boundaries of $T_t$. From the proof of Thm D.3, we can deduce that for whichever indexing of the reference set $\boldsymbol{S}$, denoting $S_1....S_n$, and for whichever equivalent boundary $\boldsymbol{M}$ produced by the algorithm, there is an ordering $M_1...M_n$ such that

$$\forall i, \boldsymbol{M_i} \equiv_T \boldsymbol{S_i} | \boldsymbol{S} \setminus \boldsymbol{S_i} \tag{5}$$

This is the case since the proof of Thm D.3 does not rely on a particular ordering of $\boldsymbol{S}$.

Let us consider the reference boundary $\boldsymbol{S}$ and any other boundary $\boldsymbol{M}$ of $T_t$. By supposition, $\boldsymbol{M}$ is included in the solutions of the algorithm. Let us consider a reference indexing $\mathcal{I}$ of $\boldsymbol{S}$ and $\boldsymbol{M}$ as is provided by the successive equivalent sets of our algorithm. Define any function $f : [|1, n|] \to 0, 1$, resulting in a intermediary set $\boldsymbol{A}$.

There is a permutation $\sigma : [|1, n|] \to [|1, n|]$ such that $f \circ \sigma$ is monotonically decreasing. We reindex $S'_i = S_{\sigma^{-1}(i)}$ and similarly for $M'_i$. Since Eq.(5) holds for any common reindexing of $\boldsymbol{S}$ and $\boldsymbol{M}$, it holds for $\boldsymbol{S'}$ and $\boldsymbol{M'}$. Also, there is an integer $k$ for which the set $\boldsymbol{M'_k} \cup (\boldsymbol{S'} \setminus \boldsymbol{S'_k})$ is identical to $\boldsymbol{A}$. By Thm.D.3, $\boldsymbol{M'_k} \cup (\boldsymbol{S'} \setminus \boldsymbol{S'_k})$ is a Markov boundary.

Direction $\impliedby$: suppose that for any correctly indexed $\boldsymbol{M}$, any function $f : [|1, n|] \to 0, 1$ induces a set $\boldsymbol{A}$ and $\boldsymbol{A}$ is a Markov boundary of $T_t$. In particular, for any $k \in [|1, n|]$, the function $f_k : i \mapsto 1$ if $k = i$ else $0$ induces a Markov boundary $\boldsymbol{S}^{-k} \cup M_k$. Therefore, we deduce that $S_k \perp\!\!\!\perp T_t | \boldsymbol{S}^{-k} \cup M_k$ for all $k$, which is exactly the conditions verified by the algorithm. Hence, each $M_k$ is included in an equivalence set, and $\boldsymbol{M}$ is an equivalent Markov boundary produced by the algorithm. □

*Residual tests (Thm.3.2).* Since $T_t \perp\!\!\!\perp X | \boldsymbol{Z}$, then $T_t, \boldsymbol{Z} \perp\!\!\!\perp X | \boldsymbol{Z}$. Any deterministic transformation of $T_t, \boldsymbol{Z}$ cannot add information so $f(T_t, \boldsymbol{Z}) \perp\!\!\!\perp X | \boldsymbol{Z}$. Using $Res \perp\!\!\!\perp \boldsymbol{Z}$, by contraction (Lemma A.2), $Res \perp\!\!\!\perp X, \boldsymbol{Z}$. Thus (decomposition) $Res \perp\!\!\!\perp X$. □

*Residual tests with composition (Thm.3.2).* Starting from $Res \perp\!\!\!\perp X, \boldsymbol{Z}$ and $Res \perp\!\!\!\perp \boldsymbol{Z}$, by weak union (Lemma A.1), $Res \perp\!\!\!\perp X | \boldsymbol{Z}$. Naturally, $Res, \boldsymbol{Z} \perp\!\!\!\perp X | \boldsymbol{Z}$. Any deterministic transformation of $Res, \boldsymbol{Z}$ cannot add information and $T_t = f(Res, \boldsymbol{Z})$. So $T_t \perp\!\!\!\perp X | \boldsymbol{Z}$. □

*Residual tests for normal variables (Thm.3.2).* Since the distribution is joint normal, then $Res = T_t - (a_0 + \sum_i a_i Z_i)$ is also a normal variable and $\boldsymbol{V} \cup \{Res\}$ is also joint normal. By lemma A.4, we can verify that this distribution verifies composition.

The residuals must be independent from $\boldsymbol{Z}$ as joint normal variables are independent if and only if they are uncorrelated. Thus, pairing $Res \perp\!\!\!\perp X$ and $Res \perp\!\!\!\perp \boldsymbol{Z}$ with composition (Assump.2.1), $Res \perp\!\!\!\perp X, \boldsymbol{Z}$. □

# E  Additional information on experiments

## E.1  Tuning study

We optimize the hyperparameters for each algorithm and forecaster we consider (see Table 3). We use the python library `optuna` with Grid Search optimization. ChronoEpilogi thresholds cover several orders of value between $10^{-20}$ and $0.05$. As we observed that for GroupLasso regularization parameter $10^{-20}$, all TS where select no matter the dataset, and at $0.1$, no TS was selected other than the target past, GroupLasso group regularization parameter ranges 25 values within this range on a logarithmic scale.

We describe the pipeline used for tuning and evaluation in Fig.3. Cross-validation on time series is done in a fixed start window growing along time. The holdout set size is consistently 1000 for synthetic MTS, and 30 percent of the MTS for real dataset. The smallest training window size must

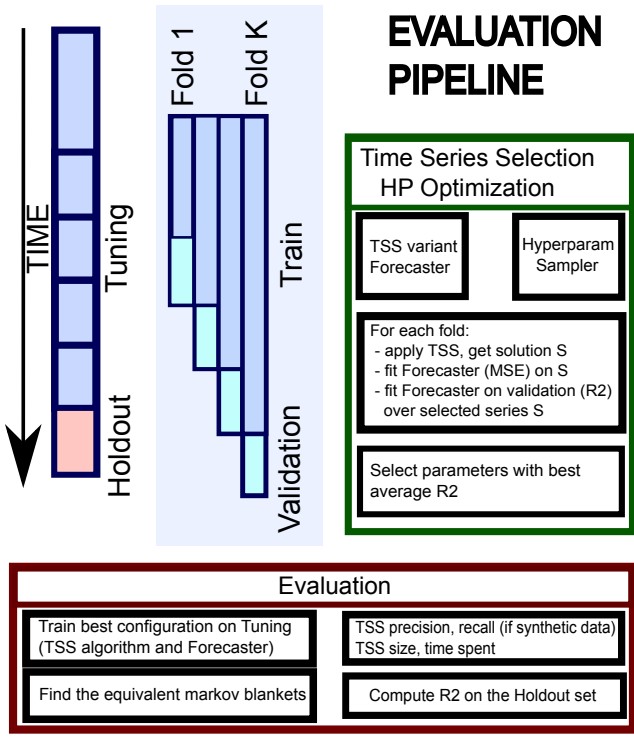

Figure 3: General experimental pipeline. We use K-fold cross validation suited to time series, for the tuning of hyperparameters of both TSS algorithms and forecasters.

Table 3: Algorithms hyperparameters

| Model | Tuned hyperparameters |
|---|---|
| GroupLasso [NST13] | group regularization coefficient |
| ARDL [PSS01] | Trend (no trend, constant, linear) |
| SVR | Kernel (rbf, sigmoid), regularization weight |
| DeepAR [SFGJ20] | Layer size, #layers, dropout rate |
| TFT [LALP21] | Layer size, #layers, dropout rate, #attention heads |

cover at least 40 percents of the Tuning split, so to obtain meaningful results and retain enough samples.

Over Fig.4, we examine how metrics vary when the thresholds are chosen to maximize the $R^2$ of the forward-backward solution $S$. We observe that the forward and backward threshold depend on the $R^2$ value, hence the proportion useful information variation compared to the total variation of the data. We confirm that different thresholds do not result in vastly different performances.

The standard deviation of the $R^2$ of the top 5 configurations for each MTS on average is 0.001 for FBE and FE compared to 0.02 for GroupLasso. See 5 for the boxplot of the distribution. This indicate a much higher impact of hyperparameter tuning on GroupLasso than ChronoEpilogi, and participates to the difficulty of tuning GroupLasso. Practitioners should expect spending more time tuning regularization-based solution than p-value-based solutions.

### E.2 Statistical significance of results on real datasets

The statistical analysis was conducted for 3 populations with 50 paired samples. The family-wise significance level of the tests is alpha=0.050. We rejected the null hypothesis that the population is normal for the populations ChronoEpilogi (p=0.000), GroupLasso (p=0.000), and NoSelection (p=0.000). Therefore, we assume that not all populations are normal. Because we have more than two populations and some of them are not normal, we use the non-parametric Friedman test as omnibus test to determine if there are any significant differences between the median values of the populations.

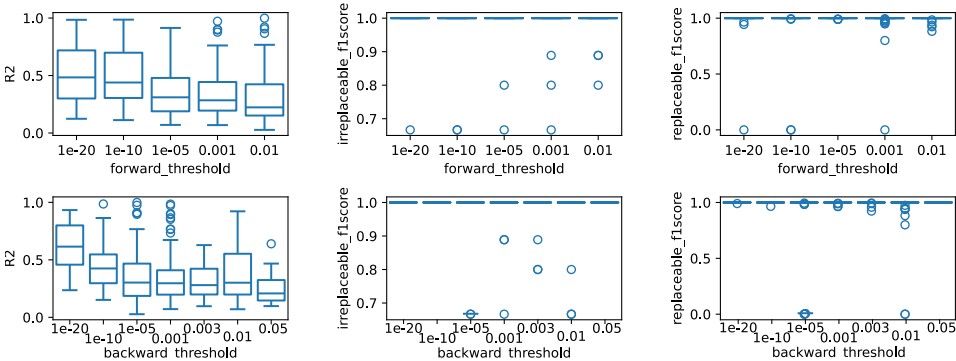

Figure 4: Optimal hyperparameters choice influence on performance metrics for version FBE. Lines corresponds to the forward phase threshold and backward phase threshold. Columns correspond to metrics (R2, f1score on the detection of irreplaceable variables, f1-score on the detection of replaceable variables.

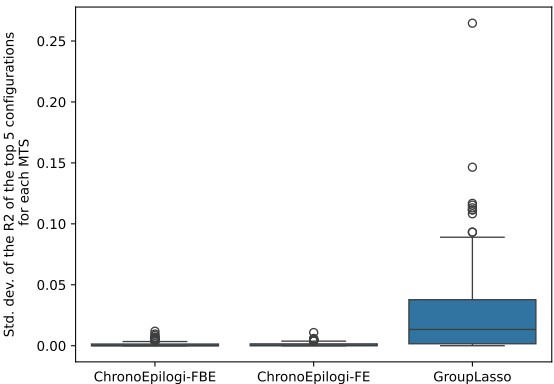

Figure 5: Standard deviation of test R2 of the top 5 configurations, over the 270 MTS of the synthetic dataset, for version FBE, FE and GroupLasso.

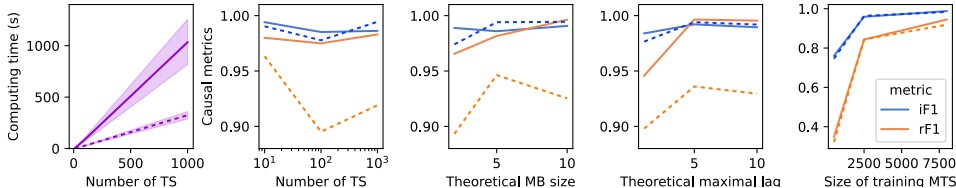

Figure 6: Evaluation of FBE (solid lines) and FE (dashed lines) over the synthetic dataset. Column 1: computing time of each method. Columns 2-5: Value of each causal metric for varying data characteristics. For columns 1-4, we report the performance of maximal sample size (8000).

Table 4: FBE predictive performance on Traffic and METR-LA. We observe that version FBE increases predictive performance compared to FE while diminishing selected size, at the cost of higher computational time

| dataset | R2 | rmse | mape | size | time | #MB |
|---------|-----|------|------|------|------|-----|
| METR-LA | 0.922622 | 0.316716 | 1.135145 | 2.1 | 1629.063206 | 2.8 |
| Traffic | 0.817231 | 0.413731 | 30.916130 | 5.3 | 14777.317261 | 36288.3 |

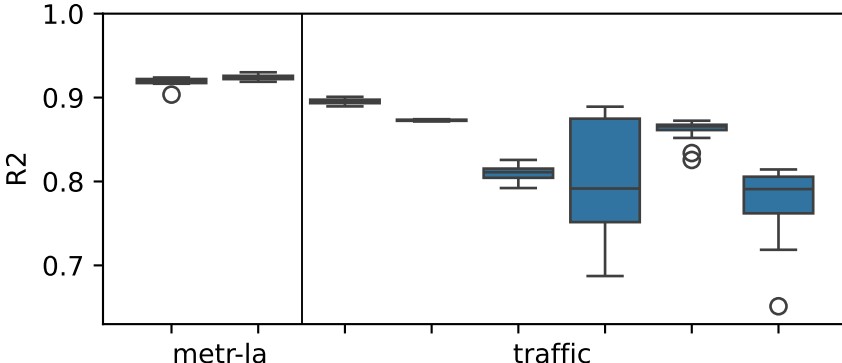

Figure 7: Multiple solutions predicitive performance over METR-LA and Traffic. When more than 20 solutions are found, we randomly select 20 of them for evaluation. We observe that both the number of targets for which there are multiple solutions decreases, with a diminution of the number of solutions overall.

We use the post-hoc Nemenyi test to infer which differences are significant. We report the median (MD), the median absolute deviation (MAD) and the mean rank (MR) among all populations over the samples. Differences between populations are significant, if the difference of the mean rank is greater than the critical distance CD=0.469 of the Nemenyi test. We reject the null hypothesis (p=0.000) of the Friedman test that there is no difference in the central tendency of the populations ChronoEpilogi (MD=0.289+-0.112, MAD=0.105, MR=2.580), GroupLasso (MD=0.287+-0.106, MAD=0.106, MR=2.320), and NoSelection (MD=0.442+-0.310, MAD=0.257, MR=1.100). Therefore, we assume that there is a statistically significant difference between the median values of the populations. Based on the post-hoc Nemenyi test, we assume that there are no significant differences within the following groups: ChronoEpilogi and GroupLasso. All other differences are significant.

### E.3   ChronoEpilogi-FBE on Traffic and METR-LA

For comparison purpose, we ran the full version of the algorithm on two of the real datasets. The performance is reported in Table 4. The number of target for which multiple solutions are found is reduced to 2 in METR-LA and 6 on Traffic (Figure 8). We observe that overall, results from synthetic datasets are verified: FBE decreases selected size and diminishes the total number of solutions, with higher computational time, resulting in an increase in predictive performance. We confirm that version FE tend to select false positives, but keep the constatation that mutliple solution exist in real datasets.

### E.4   SVR models outperforming TFT models on Solar

A surprising experimental result was that at the end of parameter tuning, SVRModels were outperforming TFT models during cross validation. As we suspected that cross validation could influence negatively the TFT performance (earlier split having less data, and Transformer-based models notoriously requiring consequent datasets), we tried a different pipeline where tuning and testing was done on a single training/validation/test split with 70,20,10 repartition. The result are in Table 5.

Table 5: Forecasting performance on Solar with modified tuning protocol. TFT models win on most targets. The higher computation time is due to running the experiment on a 16 cores machine (instead of 36) due to server availability.

| TSS | $R^2$ | rmse | mape | size | time F/E | #model | #MB |
|---|---|---|---|---|---|---|---|
| ChronoEpilogi-FE | 0.990 | 0.087 | 0.730 | 5.1 | 3/473 | 10/0/0 | 1 |
| GroupLasso | 0.989 | 0.089 | 0.600 | 4.7 | 296 | 8/0/2 | NA |
| NoSelection | 0.985 | 0.100 | 0.612 | 138.0 | NA | 10/0/0 | NA |

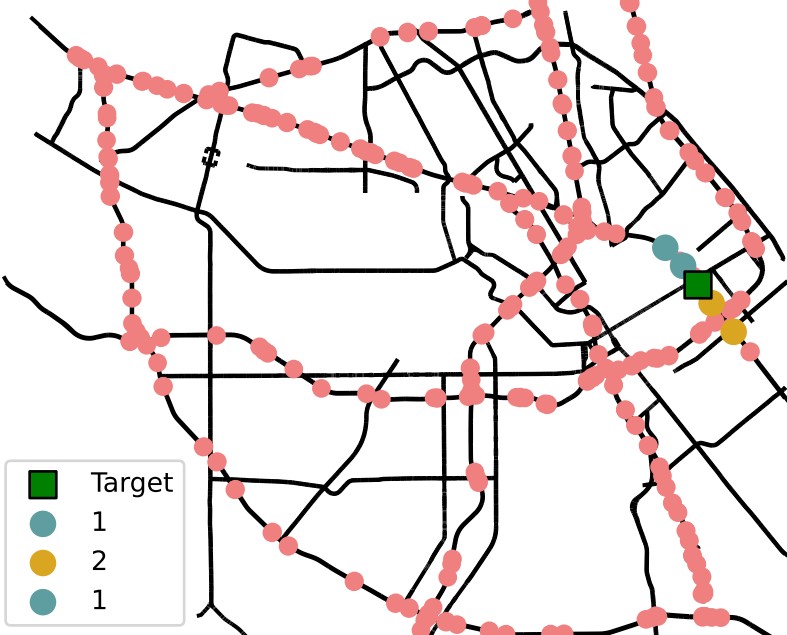

Figure 8: PEMS-BAY dataset visualisation. The sensors (dots) are overlaid with the highways and primary road networks of San Jose. Red sensors are either redundant or irrelevant. The green square figures the target variable. Blue sensors are irreplaceable and the two orange ones are replaceable sensors in a same equivalence class. We observe, confirming the observations of the original paper [LYSL18], that important predictors are close spatially.

We note that no matter the predictive model, ChronoEpilogi does not return multiple solutions on this dataset. Additionally, the results using the modified pipeline do not change the method ranking of ChronoEpilogi, GroupLasso and NoSelection.

## E.5 Feature Importance under the existence of multiple Markov Boundaries

It is known that Shapley values currently suffer from inclusion of unrealistic data instances when features are correlated, the explanations even of linear models can be misleading [AJL21]. In this section, we investigate how SHAP explanations of regression models might misrepresent the role of variables belonging to some but not all Markov boundaries of the modeled target.

To assess the impact of replaceable variables on SHAP explanations, we tune and train Support Vector Regressors (SVR) and XGBoost Regressors (XGB) for each of the 90 synthetic MTS with 100 covariates (denoted as *full MTS*), using a window of 10 timesteps. We then compute the SHAP values of random holdout samples using the KernelSHAP approximation [LL17], that we sum along the time axis to obtain the overall importance of variables (per additivity property of SHAP values [KVSF20]). We consider the average of those values as global explanations [BSC+21]. We obtain average SHAP values $s_1^i...s_{100}^i$ corresponding to the global importance of each variable in the MTS number $i$.

To evaluate the total importance for each set of equivalent variables, without the disturbance brought by the presence of replaceable variables, we apply the same average SHAP computations as previously to the 90 MTS, where we remove all but one replaceable variable per equivalent set (denoted as *reduced MTS*), to compare with SHAP explanation of unique Markov Boundary processes. For instance, if in MTS $i$ variables $\{1, 4, 16, 78\}$ are equivalent, we remove $\{4, 16, 78\}$ and compute the reduced importance $sr_1^i$ of the representative 1. Note that variable 1 is irreplaceable in the reduced MTS.

For each equivalent set $\{j_1, ..., j_c\}$ in MTS $i$, we compute the contribution of each variable $j \in \{j_1, ..., j_c\}$ to the total absolute importance of the set, as $\frac{|s_j^i|}{\sum_k |s_{j_k}^i|}$. We plot the contribution of the most important variable, second most important, etc (Fig. 9). We choose to use absolute importance values as averages are not easily comparable with relative importance. We observe that **importance is not equally shared among equivalent variables**. Models tend to select one or a few variables as important predictors. Additionally, **the importance of the top variable relative to the other diminishes as the number of equivalent variables grows**.

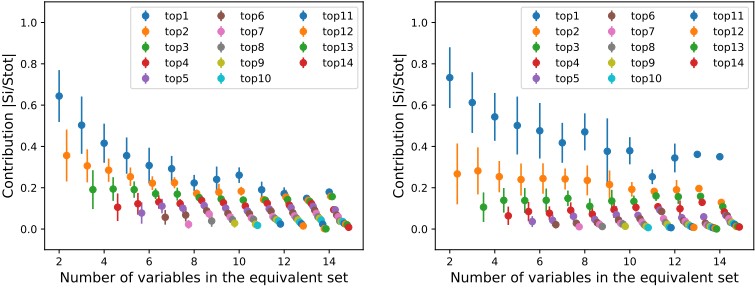

Figure 9: Average proportion of importance of top1, top2, ... variables in an equivalent set among the total importance of the equivalent set for a) SVR model b) XGB model. Models tend to focus predominantly on one variable, but with diminishing importance as the number of equivalent variables grows.

Table 6: **The most important variable of an equivalent set is highly unstable to data resampling.** For each synthetic MTS with 100 variables, the stability (against data variability) of the most important (top1) variable in each equivalent set is measured with the stability [NSB18] metric. Bootstraps are computed by subsampling contiguous intervals of timestamps, for 21 evaluations. The metric adjusts for equivalent set size. We observe that the top1 variable of an equivalent set is highly unstable. Relevance estimation based on feature stability might fail when replaceable variables exist.

| Model | Average stability (higher is more stable) | Standard deviation of stability | Theoretical min (most unstable): -1/(M-1) with M number of bootstraps | Theoretical max (most stable) |
|---|---|---|---|---|
| XGB | 0.052 | 0.110 | -0.05 | 1 |
| SVR | 0.020 | 0.089 | -0.05 | 1 |

## E.6 Hardware and Codebase

We make available our anonymized codebase at https://github.com/ev07/ChronoEpilogi.

We run our experiments on servers runing Ubuntu 22.04.4 LTS, with 36 cores, 1TB RAM, GPU Quadro RTX 8000 with driver version 550.54.14 and CUDA version 12.4.

We include in the repository a requirements file `requirements.txt` listing the necessary dependencies.

Table 7: **Presence of replaceable variable increases the instability of top important variables.** For each synthetic MTS with 100 variables, all variables but one in each equivalent sets are removed to make a reduced MTS with a unique Markov Boundary. XGB models are computed and SHAP explanations produced. We compute the stability [NSB18] of the $|MB(T)|$ most important variables in the entire explanation, for the original MTS (with multiple MB(T)) and the reduced MTS (with a unique MB(T)). Note that the metric adjusts for total number of variables in the dataset [NSB18]. We observe that explanations are significantly more unstable when replaceable variables are present.

| Model | Average Stability | Standard deviation of Stability | P-value of a paired wilcoxon signed rank test (lower indicates that XGB (reduced) is more stable) |
|---|---|---|---|
| XGB (original) | 0.380 | 0.140 | 4.2 e-13 |
| XGB (reduced) | 0.516 | 0.195 | |

