# OpenReview forum: "ChronoEpilogi: Scalable Time Series Selection with Multiple Solutions"
_NeurIPS.cc/2024/Conference — NeurIPS 2024 poster_

### Official Review · Reviewer_5YNE · 2024-07-09

**Soundness:** 2
**Presentation:** 3
**Contribution:** 2
**Rating:** 5
**Confidence:** 2

**Summary:**

The authors consider the problem of feature selection when forecasting multivariate time series. They propose a novel algorithm called ChronoEpilogi based on identifying a Markov boundary of the time series variables. They experimentally and theoretically validate the findings.

**Strengths:**

1. A significant problem to tackle,
2. Good formalization of the problem,
3. The paper is generally well-written.

**Weaknesses:**

1. Not all limitations are discussed: for instance, the model assumes that the selected set of variables remains fixed over time. When we deploy time series models, it is important that the method should work with different train and test time segments. However, since the set of variables is selected, generally, it might not be applicable to other train/test splits and thus might cause issues in real-world use.

2. In my view, the experiments could be more comprehensive. It would be beneficial to consider other forecasting models, baselines, and datasets to provide a more robust evaluation of the model's performance.

3. Prior work discussion is incomplete: for instance, signature paths and feature selection should be discussed and compared to the method. Some specific examples include
    - Cross-correlation analysis,
    - Signature transforms https://arxiv.org/abs/1603.03788

**Questions:**

1. The abstract states, “Identifying these subsets leads to gaining insights, domain intuition, and a better understanding of the data-generating mechanism.” Is this claim supported by the experiments or otherwise in the main text?
2. I have the same question about “identifying all such minimal-size, optimally predictive subsets is necessary for knowledge discovery and important to avoid misleading a practitioner.” I guess feature selection helps with interpretability, but I think the main text does not discuss it in detail.
3. Please comment on weakness #1, which I have listed above.


## Additional comments:

Additional comments
I am really surprised that the primary area is interpretability and explainability, even though the paper almost does not discuss these aspects of the solution.

What is V in tvs?

“under Composition, Interchangeability, and other broad, non-parametric assumptions” – could you list all the assumptions here?

L102 : conditional independence is not written well

L205 RVS is not properly capitalized

Eq 2: what does the dot mean? Is it a misprint?

Table 2:
What do you mean by size?
No selection yields worse results. I wonder, what happens if you use a stronger base model for forecasting, e.g., LSTM?

Re: Statistical significance:
The paper does not report statistical significance in Table 2.

**Limitations:**

In weaknesses, I’ve already commented that some crucial assumptions have not been discussed. These assumptions are critical to the framework, in my opinion.

---

> ### Author Rebuttal · Authors · 2024-08-06
>
> We thank you very much for all the constructive comments, remarks, and questions that helped us to improve the quality of our manuscript. We hope this response addresses your concerns effectively!
>
> **Q1**
> We argue that finding one MB primarily serves to build a surrogate model with improved performance. On the other hand, finding multiple MBs is essential for domain discovery and helps practitioners understand the underlying data generation mechanism used to train a model. This is about Data Explainability rather than Model Explainability and our experiments with real datasets (see Figure 1 and Table 2) provide evidence that it is an issue of practical concern. To the best of our knowledge, our submission is the first work that reports multiple MBs on popular datasets used for MTS forecasting problems.
>
> **Q2**
> The main text contains a quantitative evaluation of explanations using Markov boundaries (MBs). We show that the ChronoEpilogi algorithm reduces the number of time series to 5% of the original size, with increased performance to no selection (by 0.02 to 0.08 R2) (see Table 2 in the paper). Multiple signatures (a.k.a explanations) provide a complete set of explanations. The MBs contain the direct causes of the outcome and facilitate the interpretation of the data generation mechanism. A qualitative and subjective evaluation of the benefits for interpretation to a human user would require a user study and is out of the scope of this paper.
> A qualitative and subjective evaluation of the benefits for interpretation to a human user would require a user study and is out of the scope of this paper. We are highlighting in the paper conclusions potential usages of the multiple solutions computed by ChronoEpilogi e.g., constructing an ensemble model trained on several or all MBs as a means to create forecasters more robust to noise, faulty sensors, or other systematic errors.
>
> **W1**
> We agree that data stationarity (with a unique distribution over the entire dataset) is a limitation of our current work, as in most cause discovery algorithms (https://doi.org/10.1613/jair.1.13428). There exist indeed many scenarios where such an assumption is broken, including datasets with trends, datasets where train and tests are both stationary but do not come from the same distribution, or datasets with change points. This is an open problem in the causal discovery literature that deserves further research to identify in a first phase realistic evolution patterns of Markov Boundaries, unlike the strong hypothesis of a few previous works. Additionally, all experiments follow the Forward Chaining Cross Validation protocol. We will add these clarifications to the Camera-Ready version of the paper.
>
> **W2**
> Regarding competitors, algorithms for multiple MBs discovery assume iid data and do not provide open source code for MTS that we could include in our experiments. Their implementation is challenging as time series require specific precautions in terms of estimators since data is not iid. Moreover, the only algorithms with theoretical guarantees for multiple MB discovery are TIE* and KIAMB which require an exponential runtime and could not run for the high-dimensional TS datasets of our experimental evaluation. See also our global rebuttal answer.
>
> **W3**
> We thank you for pointing out the Signature transforms work. However, we cannot understand how we could leverage Signature Transforms  (a Feature Extraction method) to perform Feature Selection with multiple solutions in Multivariate Time Series. Getting more suggestions and pointers will be valuable and could allow us to add more discussion and material in the new paper version. We will be very willing to understand how this ML technique can be relevant to the discovery of multiple equivalent solutions (subsets of features) discovery for time series forecasting (MBo).
>
> **AC1**
> ChronoEpilogi is the first algorithm for Time-series Variable Selection with Multiple Solutions that opens new perspectives in Data Explainability compared to Model Explainability heavily studied in the xAI literature. We argue that existing feature importance methods like SHAP are challenged by the presence of multiple MB in the datasets. This is a new claim that we are planning to support with additional experiments that will appear in the camera-ready version. It is known that Shapley values currently suffer from the inclusion of unrealistic data instances when features are correlated, even for linear models. We are currently investigating how SHAP explanations of regression models might misrepresent the role of variables belonging to some but not all Markov boundaries of the modeled target. We claim that the SHAP importance score of each equivalent set of variables is distributed among equivalent variables, hence leading to underestimations of an equivalent set importance, when considered individually. This attribution is unstable: on different data splits, different variables among an equivalence set obtain high importance. See global review. To the best of our knowledge, these results have not been reported in the existing xAI literature.
>
> **AC7**
> For each MTS, we obtain the size of the solutions found by GroupLasso and ChronoEpilogi (all solutions produced by ChronoEpilogi are the same size) by computing their mean and standard deviation.
>
> **AC8**
> We evaluated ChonoEpilogi and its competitors with experiments conducted over three real TS datasets. For each one, we test 10 different targets. We are currently experimenting with additional real datasets fitting multiple Markov Boundaries (MB) that allow us to reliably compute a critical difference diagram of the statistically significant ranking of GroupLasso and ChronoEpilogi performance using the post-hoc Nemenyi test. In the new TS datasets, results are similar to the ones shown so far and strengthen our claims.
>
> The typos have been corrected.

---

> > ### Comment · Reviewer_5YNE · 2024-08-09
> >
> > Thank you for the response!
> >
> > The response clarifies my concern. I think this is a very interesting aspect of your work about data explainability. In my opinion, viewing this work from this perspective would make it a really valuable contribution.
> >
> > Currently, my concern is that adding such a massive claim would make the paper significantly different from the version under review. Could you briefly list all the changes you plan to incorporate?

---

> ### Author Response · Authors · 2024-08-12
> **Additional Contributions**
>
> We thank you for your positive feedback on our answer during the rebuttal.
>
> We will definitively further detail domain discovery under the light of the Data Explainability in the paper introduction, add new claims regarding the limitation of Feature Importance methods in the presence of redundant features and include in the main paper and the appendix the experiments that support them.
>
> We should stress that that this a fourth contribution of our work that could not been made without the previous three :
>
> (a) ChronoEpilogi algorithm variants
>
> (b) Theoretical guarantees of the algorithm
>
> (c) quantitative evaluation of ChronoEpilogi results with synthetic and real datasets.
>
> More precisely, we will make the following new claims, based on the SHAP value experiments we have included in the rebuttal:
>
> Claim 7: The importance score of variable with equivalences decreases by half in our experiments, as the number of equivalent variables grows.
>
> Claim 8: the top |MB(T)| predictors in explanations are significantly more unstable to data changes in presence of multiple Markov Boundaries compared to distributions with only one MB(T) (pvalue 4e-13).
>
> The above claims can be validated from the one pdf  page we added in our global answer. The experimentation protocol (models libraries, SHAP explainer implementation, tuning, instability metric) we used will be also detailed.
>
> We believe that, on the additional page of the camera-ready version we will be able to include the main results of our new experiments in the paper beside the appendix.

---

> > ### Comment · Reviewer_5YNE · 2024-08-14
> >
> > Thank you for responding! Your clarifications helped me much better understand the contribution. I will raise my score to 5.

---

> > > ### Author Response · Authors · 2024-08-14
> > >
> > > Dear reviewer, thank you a lot for the time you have spent on our paper and for the constructive feedback. We believe this work will be valuable for researchers of our domain.

---

### Official Review · Reviewer_fxYh · 2024-07-12

**Soundness:** 3
**Presentation:** 3
**Contribution:** 3
**Rating:** 7
**Confidence:** 3

**Summary:**

The authors propose a scalable algorithm called ChronoEpilogi that aims to select multiple subsets (Markov Boundaries) of time series (TS) features in order to better understand the underlying data generation process and to provide better explanations of downstream forecasting tasks. Through extensive experiments, the authors show that time series forecasting models perform better when fed with these subsets of TS features (individually) than when fed with all TS features.

**Strengths:**

-  Originality: Although the problem addressed in this paper is not new and the proposed solution is based on the combination of existing methods/models, the originality lies in the fact that, unlike previous work, the proposed algorithms provide a compact representation of mutually equivalent variables for multivariate time series (MTS). In addition, the redundant and irreplaceable variables in MTS can be relatively easy to identify.

- Quality: The document is well structured and the assertions are fairly well supported.

- Clarity: Overall, the document is well written and pleasant to read. However, the reviewer suggests improving definition 2. What does V stand for?

- Significance: The reviewer believes the proposed algorithm can be used as an alternative solution to provide explainability in diverse and sensitive fields such as medicine and autonomous vehicles.  Indeed, in these fields, when it comes to forecasting tasks, identifying the  the time series  features that influence the decision is as important as model accuracy.

**Weaknesses:**

- The experiment is not conducted with multivariate time series that have a high rate of missing values. This is a crucial aspect that the study should have taken into account, as missing values are inherent in time series and may affect causal inference.;

- The authors simply identify subsets of time series variables without providing concrete explanations that could have strengthened their claim. For example, the authors should have taken a few subsets (Markov Boundaries)  in any task and shown how they are actually relevant to the target;

- The reviewer understands the usefulness of greedy heuristics to speed up the algorithm. However, this heuristic has the disadvantage of providing suboptimal results. Although the authors demonstrate its effectiveness, it would be interesting in future work to test it on additional datasets covering different domains.

**Questions:**

- Why did you choose ARDL over other models like RNNs which are also autoregressive models and are certainly more efficient (Equation 2)?

- Is the model redesigned at each iteration of the Forward phase, or does it remain unchanged? The reviewer asks this question because the size  of the inputs $\textbf{S}$ may vary at each iteration (see Algorithm 3 line 8)?

- What can explain the outperformance of SVR over TFT and DeepAR with the Solar Dataset?

- How do the authors expect their algorithm to perform when faced with sparse multivariate time series, given that missing value can affect causal inference?

- The reviewer does not understand the relevance of reporting the standard deviation in columns MB size and Number of MB (Table 1). Could you please elaborate on this?

**Limitations:**

- The authors have identified the limitations of their work and plan to address them in their future work;

- On line 176 it should be forward instead of backward;

-  The reviewer suggests improving definition 2, which is the core definition of the paper.

---

> ### Author Rebuttal · Authors · 2024-08-06
>
> We thank you very much for all the constructive comments, remarks, and questions that helped us to improve the quality of our manuscript. Hereafter, we provide answers to all your questions and comments pointed out in the weaknesses and limitations sections of your review.
>
> **Question 1)**
>
> In a preliminary phase of experimentation, we indeed planned to include LSTM models to tackle nonlinear tasks. Our observation was that applying the forward backward phases to find an MB(T) in nonlinear synthetic datasets (https://doi.org/10.1613/jair.1.13428) had poor causal recovery when using LSTMs compared to linear methods. We explained this by the variability of DNN training which has higher fluctuations in R2 compared to the increase in R2 expected due to adding a correct MB(T) variable. Obtaining sufficient stability might require more data, finer human analysis of model training routines, and extensive bootstrapping to remove fluctuations. Therefore, we did not include LSTM in ChronoEpilogi.
>
> **Question 2)**
>
> At every iteration, we build a new model with the variables selected at that step. It is important to note that ChronoEpilogi avoids the creation of a new model for every possible variable set by examining the correlation of the model residuals with the variable set selected at the previous step.
>
> **Question 3)**
>
> The discrepancy is likely to be influenced by cross-validation, where the training set in the earliest split is 50% smaller than the one of the last split. SVR results are stable across folds, while TFT models have more unstable results. We plan to add, in the appendix, new results for this experiment with a different cross-validation strategy to confirm our hypothesis.
>
>
> **Question 4)**
>
> ChronoEpilogi requires two main operators: (a) a modeler, and (b) an estimator of pairwise correlation between the residuals and a Time Series variable. One could impute missing values before constructing a model or computing correlation (https://folia.unifr.ch/unifr/documents/309429). Alternatively, one could employ models that inherently fit models in the presence of missing values (https://doi.org/10.1038/s41598-018-24271-9). The absolute performance of such a version of our algorithm, as well as its relative performance against similar types of algorithms on datasets with missing values, is an open research question for causal discovery methods (https://doi.org/10.1613/jair.1.13428). In this paper, we propose the first algorithm for Time Series Selection with Multiple Solutions, so that we can extend it in the future to handle datasets with missing values.
>
> **Question 5)**
>
> In Table 1, we observe that ChronoEpilogi discovers (on average) smaller solutions (with smaller variance) than GL. Standard deviation and mean are computed across the 270 synthetic MTS, where for each MTS ChronoEpilogi and GL are applied once. For each MTS, we obtain the size of the solutions found by GroupLasso and ChronoEpilogi (all solutions produced by ChronoEpilogi are the same size) by computing their mean and standard deviation.
> The standard deviation of  MB size is relevant to our experiment, as it allows us to understand that GroupLasso computes feature sets with varying lengths compared to those of ChronoEpilogi. GL tends to include too many variables generating overfitting in linear models.
> We also observe that GL tends to dismiss some irreplaceable variables that should have a low impact on predictive performance.
> Evaluating the standard deviation of solution size is also relevant to compare FBE and FE (approximate version). We observe that FE tends to create more solutions than FBE, with higher fluctuations. As reported in Table 2 (real datasets), both methods can provide parsimonious solutions.  To better appreciate the quality of the approximation offered by FE we will include in the appendix also the characteristics of the solutions computed by FBE in real datasets.
>
> **Weakness 1)**
>
> We consider this as an important issue, but it is orthogonal to our current submission also as in recent empirical studies of causal discovery methods (https://doi.org/10.1613/jair.1.13428).
>
> **Weakness 2)**
>
> We started investigating selected subsets in real TS datasets as Traffic (see Figure 1).  This study is interesting to reveal the reasons why multiple replaceable TS variables may exist, as for instance the spatial proximity of traffic sensors included in the dataset.  However, such root cause analysis could not be completed without the involvement of experts from different domains, a task that is not possible with the public TS datasets used in our experimental evaluation.  We are planning in the future to conduct a deeper analysis of proprietary datasets in the context of an ongoing research project. As noted in the paper conclusions, the discovery of multiple MBs aims to facilitate a better understanding (via appropriate visualization tools) of the underlying data generation mechanism, as well as the construction of ensemble models trained on several or all MBs as a means to create forecasters more robust to noise, faulty sensors, or other systematic errors.
>
> **Weakness 3)**
>
> Current MTS datasets commonly used for forecasting do suffer from lack of domain diversity (https://arxiv.org/pdf/2310.06119) (https://arxiv.org/html/2403.20150v1). We are considering datasets from other domains (such as maintenance and water flow forecasting), but those are proprietary and outside of the scope of this paper due to each dataset's specific challenges.
> We are currently experimenting with two more additional datasets: METR-LA and PEMS-BAY (speed readings in road networks) that we found to exhibit multiple MBs. The new results will be included in Table 2 and the appendix of the camera-ready version.
>
>
> **Limitation 2)**
> Fixed
>
> **Limitation 3)**
>
> We have improved definitions 2.1 and 2.3, which we include in the Camera-Ready version of the paper.

---

> > ### Comment · Reviewer_fxYh · 2024-08-12
> >
> > Dear authors, thank you for your detailed answers. My concerns were clarified. I keep the score unchanged.

---

### Official Review · Reviewer_UZ5R · 2024-07-12

**Soundness:** 3
**Presentation:** 3
**Contribution:** 2
**Rating:** 6
**Confidence:** 3

**Summary:**

The authors presents **ChronoEpilogi**, an algorithm for multiple feature selection in multivariate time-series (TS) forecasting. This approach aims to identify all minimal-size subsets of TS variables (Markov Boundaries) that optimally predict a given target variable's future. The key contributions are:

1. **Theory Development**: Introduces the problem of multiple time-series variable selection (MTVS) and the concepts of informational equivalence and interchangeability in TS data.
2. **Algorithm Design**: Proposes ChronoEpilogi, which identifies all Markov Boundaries under broad, non-parametric assumptions.
3. **Experiments and Results**: Demonstrates ChronoEpilogi's scalability to hundreds of TS variables and its effectiveness in reducing the number of variables while maintaining or improving forecasting performance.

**Strengths:**

Some of the key strengths of the paper are:

1. The paper proposes a novel theoretical foundation for multiple feature selection in TS data including concepts like informational equivalence and interchangeability. Combining these concepts the authors have been able to propose an empirical method to detect Markov Boundaries
2. Furthermore, the proposed algorithm ChronoEpilogi is shown to handle large datasets effectively, making it suitable for real-world applications with numerous TS variables. This scalability is crucial to real world usage of the algorithm
3. Another key contribution for the paper is that ChronoEpilogi aims to identify all minimal-size subsets, offering multiple valid forecasting models and insights.

**Weaknesses:**

While being a very interesting paper, there are some avenues for improvement:
1. While the authors discuss the scalability, the algorithm's computational complexity seems to be high for very large datasets, potentially limiting its practicality.
2. Some of at the assumptions that the algorithm relies on such as Compositionality and Interchangeability may not hold in all real-world scenarios, potentially affecting its generalizability. The authors should consider discussing the limitations in their papers and how well the assumptions hold in practice
3.The authors have provided thorough experimentations. However, to justify the practicality of the algorithm, it would be interesting to report additional validation on more diverse and complex real-world datasets

**Questions:**

It would be great if the authors can discuss about the computational complexity and the limitations stemming from their assumptions

**Limitations:**

See weakness above

---

> ### Author Rebuttal · Authors · 2024-08-06
>
> We thank you very much for all the constructive comments, remarks, and questions that helped us to improve the quality of our manuscript. Hereafter, we provide answers to all your questions and comments pointed out in the weaknesses and limitations sections of your review.
>
> **Question 1) It would be great if the authors can discuss about the computational complexity and the limitations stemming from their assumptions.**
> **Weakness 1) While the authors discuss the scalability, the algorithm's computational complexity seems to be high for very large datasets, potentially limiting its practicality.**
>
> ChronoEpilogi complexity essentially depends on the number and the complexity of tests run by each phase of the algorithm. Note that a model is constructed only at the end of each iteration step and not for each candidate subset of TS variables. The exact version of ChronoEpilogi, termed FBE, is indeed expensive (see experiments with synthetic data in Figure 2) but allows us to prove soundness and completeness theorems. For this reason, we have also proposed a greedy approximated version termed FE that avoids running the backward phase and computes the equivalence directly in the forward phase, thus pruning new model generations for all the selected TS variables discovered in the forward phase. As we can see in Table 1, in synthetic datasets fitting multiple MBs, FE runs in 30% of FBE time on average, with only a decrease of causal f1 metric of 0.05 while in real TS datasets, FE is able to find the first MB(T) up to 2 orders of magnitude faster than GroupLasso. Such results suggest the practicality of the proposed algorithm.
>
> **Weakness 2) Some of the assumptions that the algorithm relies on such as Compositionality and Interchangeability may not hold in all real-world scenarios, potentially affecting its generalizability. The authors should consider discussing the limitations in their papers and how well the assumptions hold in practice**
>
> Please note that without any assumption regarding the joint probability distribution, causal variable selection is NP-hard even for linear regression problems (https://www.tandfonline.com/doi/abs/10.1080/00949658208810560). Rather than simply dropping the strong Faithfulness assumption we have considered weaker assumptions like Compositionality and Interchangeability allowing us to prove ChronoEpilogi soundness and completeness.  Several practical and useful data distributions satisfy Composability.  To that extent, in example 2.1, we show that any deterministic transformation satisfies composition.  In example 2.2, we show that Joint Gaussian Distributions also satisfy Composability.  We will also include in the appendix examples of distributions that do not respect Compositionality, starting with the typical example of non-faithful XOR operation between Bernouilli variables with p=0.5.  ChronoEpilogi (and its variants) is an effective tool that allows us to experimentally verify the Compositionality and Interchangeability properties in real TS datasets. To the best of our knowledge, our submission is the first work that reports multiple MBs on popular datasets used for MTS forecasting problems related to these two weak assumptions of our algorithm.
>
> **Weakness 3.The authors have provided thorough experimentations. However, to justify the practicality of the algorithm, it would be interesting to report additional validation on more diverse and complex real-world datasets**
>
> We are currently extending our experimental study to two new TS datasets: METR-LA and PEMS-BAY (speed reading in road networks) that we found to exhibit multiple MBs (https://openreview.net/forum?id=SJiHXGWAZ). Current MTS datasets commonly used for forecasting do suffer from lack of domain diversity (https://arxiv.org/pdf/2310.06119) (https://arxiv.org/html/2403.20150v1). Please note that the challenge of the experimental evaluation of our work is not to measure forecasting performance with different models but the quantitative evaluation of ChronoEpilogi's multiple solutions against our main competitor namely GroupLasso. The newly produced results also confirm ChronoEpilogi's excellence and they will be included in Table 2 and the appendix of our paper.

---

> > ### Comment · Reviewer_UZ5R · 2024-08-09
> > **Acknowledging author response**
> >
> > Thanks for the detailed response

---

### Official Review · Reviewer_PVJ5 · 2024-07-16

**Soundness:** 3
**Presentation:** 4
**Contribution:** 3
**Rating:** 7
**Confidence:** 3

**Summary:**

This paper handles the problem of selecting all the minimal-size subsets of multivariate time series variables such that the past leads to an optimal predictive model for the forecast of a given target variable, which is essentially a time series feature selection problem. Past algorithms have worked to select a single such subset. The proposed algorithm is relatively efficient, in that it does not take as much longer than finding a single subset as one would think, but leading to more insight and better "Markov blankets."

**Strengths:**

1. The paper handles an important problem in a clever way and is explained quite clearly.
2. The experimental results are convincing and actually include running time, which is often omitted.
3. The theoretical results look correct, although admittedly I did not comb through the proofs in great detail.

**Weaknesses:**

1. The proposed algorithm was only compared against GroupLasso and not against any other among the related work mentioned in the paper.

**Questions:**

1. Line 176 should say "forward" rather than "backward."
2. I suspect that algorithm 3 step 9 and algorithm 4 step 3 should have $\geq$ in place of $\leq$.
3. In line 260, how is TSS defined?

**Limitations:**

Yes

---

> ### Author Rebuttal · Authors · 2024-08-06
>
> We thank you very much for all the constructive comments, remarks, and questions that helped us to improve the quality of our manuscript. Hereafter, we provide answers to all your questions and comments pointed out in the weaknesses and limitations sections of your review.
>
> **Question 1) Line 176 should say "forward" rather than "backward."**
>
> Thank you for finding this typo, we have fixed it in the paper.
>
> **Question 2) I suspect that algorithm 3 step 9 and algorithm 4 step 3 should have ≥ in place of ≤.**
>
> This one is another typo since we stop when we fail to reject the hypothesis stating the models are different. The smaller the p-value, the more different the tested models. We corrected it.
>
> **Question 3) In line 260, how is TSS defined?**
>
> TSS stands for Time Series variable Selection and will repeat the definition of the acronym at its first appearance in the introduction.
>
> **Weakness 1) The proposed algorithm was only compared against GroupLasso and not against any other among the related work mentioned in the paper.**
>
> This is due to the fact that competitor solutions for multiple MBs discovery assume for iid data hence do not provide open-source code for MTS data, and thus their implementation is challenging and a research work in itself. We should note that time series require specific precautions in terms of estimators since data is not iid. Moreover, the only algorithms with theoretical guarantees for multiple MB discovery are TIE* and KIAMB which require an exponential runtime (see related work) and could not run for the high-dimensional TS datasets of our experimental evaluation.

---

### Official Review · Reviewer_hgHX · 2024-07-23

**Soundness:** 3
**Presentation:** 3
**Contribution:** 3
**Rating:** 6
**Confidence:** 3

**Summary:**

This paper considers the problem of finding all minimal subsets of variables for optimal prediction of time series data, coining the term "Markov Boundaries" for those minimal subsets constituting Markov Blankets for the target time series variables in question.
The paper then proposes novel algorithms for this problem,FBE and FE, and prove the soundness and completeness of FBE (FE is a faster approximate algorithm) and empirically evaluate their performance.
The experiments are conducted using both synthetic data (with ground truth causal structure) and real world data, and compare the performance of the proposed algorithms against baselines of Group Lasso (GL) and No variable selection, with respect metrics including predictive accuracy, accuracy of causal structure learning (for synthetic data), computation time and solution size.
The results presented validate a number of claims about the proposed methods, the notable ones being that they are more accurate at uncovering the ground truth causal structure on synthetic data and FE roughly comparable to GL on real world data sets in terms of accuracy and computation time, sometimes significantly out-performing it in terms of solution size.
The problem formulating is apparently novel and interesting, and the proposed methods are also novel and theoretically sound (and complete). The empirical results show that they are at least competitive to the standard baselines.
This work would add some valuable knowledge and insights to the community with interest in causal modeling and interpretable learning in time series data.

**Strengths:**

The problem formulation is novel and interesting and well motivated practically.
The proposed solution is novel and sound and complete.
The empirical evaluation is reasonable.

**Weaknesses:**

The performance of the proposed methods against the baseline of Group Lasso on real world data sets is not exactly compelling.
More clarify on the relative advantage of the proposed method(s) would be valuable.
The optimal algorithm, FBE, is not evaluated on real world data sets, which I assume is due to computational complexity. It would be beneficial to know if any evaluation (even if partial) could be performed on FBE on the real world data.

**Questions:**

One wonders if there are ways to use Group Lasso to obtain multiple solutions of the type obtained by the proposed methods, for example, by performing multiple randomized runs with perturbation and aggregating the outputs. A comparison with such heuristics would be of interest.

**Limitations:**

The authors do mention that the claims are limited to the scope of their empirical evaluation which could be enhanced.
It would be additionally desirable to address the question of quantifying the statistical confidence of the outputs of the algorithms.

---

> ### Author Rebuttal · Authors · 2024-08-06
>
> We thank you very much for all the constructive comments, remarks, and questions that helped us to improve the quality of our manuscript. Hereafter, we provide answers to all your questions and comments pointed out in the weaknesses and limitations sections of your review.
>
> **Question 1) One wonders if there are ways to use Group Lasso to obtain multiple solutions of the type obtained by the proposed methods, for example, by performing multiple randomized runs with perturbation and aggregating the outputs. A comparison with such heuristics would be of interest.**
>
> One could extend Lasso-type algorithms based on global optimization to identify multiple solutions. An example of preliminary work in this direction or cross-sectional i.i.d. data is at https://arxiv.org/abs/1710.04995. However, how to extend this or similar types of algorithms for the Group Lasso and time-series data is still an open problem. In addition, proving soundness and completeness for algorithms based on Lasso is also challenging.
>
> **Limitation 1) The authors do mention that the claims are limited to the scope of their empirical evaluation which could be enhanced. It would be additionally desirable to address the question of quantifying the statistical confidence of the outputs of the algorithms.**
>
>
> In the current version of the paper, we evaluate ChonoEpilogi and its competitors with experiments conducted over three real TS datasets. For each one, we test 10 different targets.  This dataset number does not allow us to apply a significant statistical test on the algorithms’ predictive performance. We are currently experimenting with additional real TS datasets fitting multiple Markov Boundaries (MB) that allow us to compute a critical difference diagram of the statistically significant ranking of GroupLasso and ChronoEpilogi performance using the post-hoc Nemenyi test.
> Moreover, to assess the stability of Feature Importance methods like SHAP in the presence of replaceable variables, we are currently experimenting with synthetic datasets of varying redundancy degrees using the Wilcoxon signed rank tests. In the new datasets, results are similar to the ones shown so far and strengthen our claims.

---

> > ### Comment · Reviewer_hgHX · 2024-08-07
> > **Response regarding additional evaluation**
> >
> > Thank you for elaborating on your on-going efforts on additional empirical evaluation. It would be good (and strengthen the paper)  to include the results of those (regarding both Group Lasso and SHAP) in an updated version of the paper.

---

### Author Rebuttal · Authors · 2024-08-06

We thank all the reviewers for their constructive comments, remarks, and questions that helped us to improve the quality of our manuscript. Hereafter, we provide a summary of the main modifications and additions we will bring to the final version of the paper.

**1) Extension of the empirical datasets and consideration of other forecasting methods and baselines.**

In terms of additional datasets, we are currently extending our experimental study to two new TS datasets (speed reading in road networks), namely, METR-LA and PEMS-BAY (https://arxiv.org/abs/1707.01926) that we found to exhibit multiple MBs. With 5 datasets in total and 10 targets modeled each, we can compute critical difference diagrams of the statistically significant ranking of GroupLasso, ChronoEpilogi, and NoSelection performance. The newly produced results also confirm ChronoEpilogi's excellence and will be included in Table 2 of the camera-ready version of the paper. Note that current MTS datasets commonly used for forecasting do suffer from a lack of domain diversity (https://arxiv.org/pdf/2310.06119) (https://arxiv.org/html/2403.20150v1).
Regarding additional forecasting models, we could include Nhits and DecoderMLP available in the dedicated library we are using (https://pytorch-forecasting.readthedocs.io/en/stable/models.html). Please note that the challenge of the experimental evaluation of our work is not to measure forecasting performance with different models but the quantitative evaluation of ChronoEpilogi's multiple solutions against our main competitor namely GroupLasso.
Unfortunately, comparison with additional baselines is not possible as competitor solutions for multiple MBs discovery assume iid data and how to adapt the methods for MTS data would require a paper in itself.
We will also provide FBE results on real datasets where possible.

**2) Further evaluation of ChronoEpilogi relevance to xAi.**

ChronoEpilogi is the first algorithm for Time Series Selection with Multiple Solutions and opens new perspectives in Data Explainability compared to Model Explainability which has been heavily studied in the xAI literature.  We argue that existing feature importance methods like SHAP are challenged by the presence of multiple MB in the TS datasets.  This is a new claim that we are planning to support in the camera-ready version of the paper and we submitted as a pdf file.  It is known that Shapley values currently suffer from the inclusion of unrealistic data instances when features are correlated, even for linear models (https://www.sciencedirect.com/science/article/pii/S0004370221000539). We are currently investigating how SHAP explanations of regression models might misrepresent the role of variables belonging to some but not all Markov boundaries of the modeled target. We claim that the SHAP importance score of each equivalent set of variables is distributed among equivalent variables, hence leading to underestimations of an equivalent set importance, when considered individually. This attribution is unstable: on different data splits, different variables among an equivalence set obtain high importance. The importance of the top variable diminishes as the number of equivalent variables grows, sharing more and more importance with the remaining variables. We report the results of these experiments on the page attached to the global rebuttal. To the best of our knowledge, these results have not been reported in the existing xAI literature.

---

### Decision · Program_Chairs · 2024-09-25

**Decision:**

Accept (poster)

**Comment:**

Five reviewers all recommend acceptance. The reviews are detailed, and the authors have responded carefully. The research is thorough and novel.

This paper is somewhat interdisciplinary, because it is about feature selection in the unusual context where the features are entire time series, and with explainability as a main motivation, and therefore it likely deserves acceptance. This paper was considered for a spotlight, but due to the extremely limited space for spotlights, and when considering all of the potential spotlights across the entire NeurIPS accepted paper set, and ranking them, this paper will be instead be considered as a poster presentation. We do hope the authors will present this original interdisciplinarity as such to the audience, as making the motivation and contribution of the work easy to understand will be important, so please consider this to the extent possible.